# SELECTIVE PROMPT ANCHORING FOR CODE GENERATION

## ABSTRACT

Recent advances in large language models (LLMs) have transformed software development by automatically generating code based on users' requests in natural language. Despite these advancements, challenges remain in generating fully correct code and aligning with user intent. Our empirical study reveals LLMs tend to dilute their self-attentions on the initial prompt as more code tokens are generated. We hypothesize this self-attention dilution issue is one of the root causes of inaccuracies in LLM-generated code. To mitigate this issue, we propose **S**elective **P**rompt **A**nchoring (SPA) to amplify the influence of the selected parts in the initial prompt, which we refer to as "anchored text", during code generation. Specifically, SPA calculates the logit distribution difference with and without the anchored text. We prove this logit difference approximates the anchored text's contextual contribution to the output logits. SPA creates an augmented logit distribution by linearly combining the original logit distribution and the logit difference. We evaluate SPA with five LLMs on four benchmarks. Our results show that after tuning on a few dozen tasks, SPA consistently improves Pass@1 on new tasks by up to 7.6% across all settings. Notably, with selective text anchoring, a small version of DeepSeek-Coder (6.7B) can achieve better performance than an original much larger version (33B). Our code is available at `https://anonymous.4open.science/r/Selective-Prompt-Anchoring-74E7`.

## 1 INTRODUCTION

Large language models (LLMs) have emerged as powerful programming assistants. They have demonstrated unprecedented capabilities in interpreting natural language descriptions and generating source code. Despite this great progress, LLMs still produce incorrect solutions to some tasks or generate code that does not fully meet user expectations. The prevalence of such generation errors undermines their reliability and limits their utility in real-world software development.

To improve the performance of LLMs on coding tasks, many efforts have been made to develop high-quality training data (Li et al., 2023c; Guo et al., 2024; Wei et al., 2023) and design new domain-specific training objectives (Niu et al., 2022; Chakraborty et al., 2022). However, these approaches require tremendous computational resources. Training-free approaches have been explored to address this challenge by enhancing the prompting method or incorporating external knowledge, such as retrieval-augmented generation (Du et al., 2024), chain-of-thoughts (Le et al., 2024; Suzgun et al., 2022), self-planning and debugging (Jiang et al., 2023; Chen et al., 2023), etc. While they have been proven to be effective in improving performance, there exist limitations such as being sensitive to the quality of prompt design and retrieved data (Zhao et al., 2021). Compared with existing methods, this work aims to study and improve LLMs in an orthogonal direction through attention adjustment.

One key component of existing LLMs is the self-attention mechanism in the transformer architecture (Vaswani et al., 2017), which enables models to focus on crucial parts of the given prompt. Despite the success of the self-attention mechanism, prior works found language models exhibit simple attention patterns (Raganato & Tiedemann, 2018; Voita et al., 2019). Furthermore, an empirical study (Kou et al., 2024) found that given a coding task, there often exists a misalignment between LLM attention and human attention. Compared with human programmers, LLMs often focus on different parts of a natural language description when generating code. Inspired by this finding, we hypothesize that a root cause of inaccuracy in LLM-generated code stems from the suboptimal model

attention. To verify our hypothesis, we conduct an empirical study that analyzes the shift in LLMs' attention distribution during code generation. We observe that LLMs' attention to the initial prompt gradually dilutes as generating more code. We call this phenomenon "*attention dilution*" in code generation tasks.

In standard decoding algorithms, LLMs calculate a conditional probability for the next token based on the preceding context. However, the autoregressive nature of LLMs considers both the initial prompt and possibly wrong self-generated tokens together as the correct context and pays comparable attention to them. We argue LLMs should pay more attention to the absolutely correct prompt and less attention to the following self-generated content that could potentially be wrong.

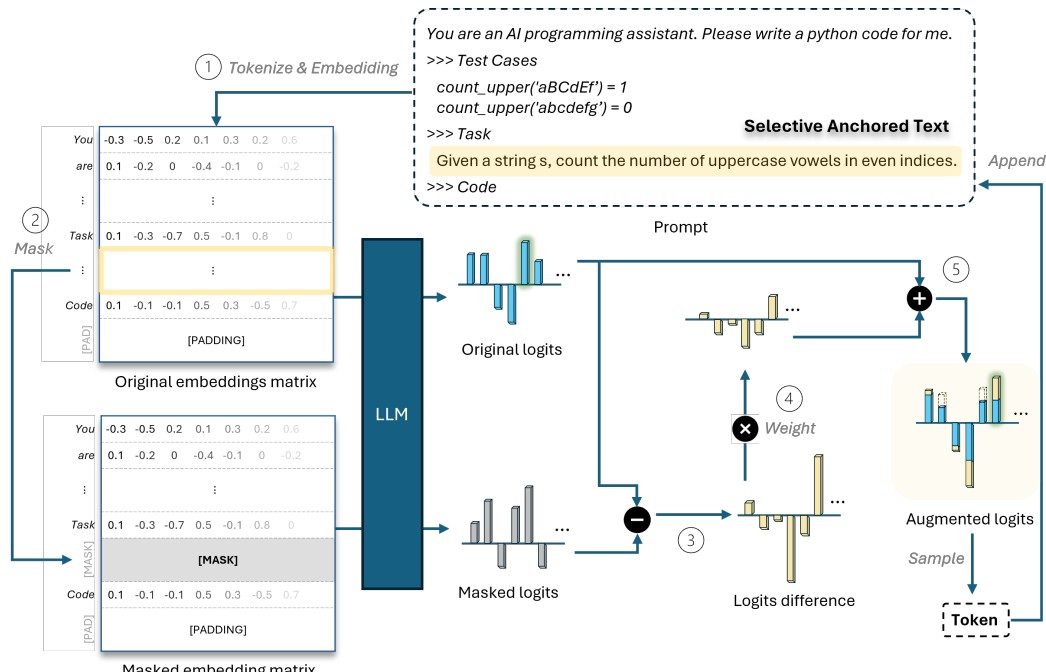

Figure 1: The Workflow of Selective Prompt Anchoring (SPA)

To mitigate this limitation, we propose **S**elective **P**rompt **A**nchoring (SPA), a model-agnostic approach that optimizes LLMs' attention by amplifying the contextual influence of selective prompt, towards each generated token. SPA is inspired by the anchoring effect (Furnham & Boo, 2011) in psychology, which refers to people being influenced by specific information given before decision-making. In SPA, we refer to this information as *anchored text*, a group of selected tokens within the prompt that should receive higher attention from the model than others.

Figure 1 illustrates the pipeline of SPA. Given the anchored text, SPA creates an original embedding matrix (①) as well as a masked embedding matrix by replacing the embeddings corresponding to anchored text with mask embeddings (②). We mathematically show that the anchored text's contextual influence can be approximately measured by the difference between the logit distribution generated from the original prompt and the prompt with the anchored text masked (③). To amplify the influence of anchored text in the model output, SPA multiplies this logit distribution difference by a hyperparameter called anchoring strength (④), and then adds it to the original logit distribution (⑤). We find while the optimal anchoring strength varies across different models and tasks, it can be easily tuned through dozens of tasks.

We evaluate SPA on four benchmarks with five LLMs. The result shows SPA can significantly and consistently boost Pass@1 across all models and benchmarks, highlighting a new direction for controlling LLMs' high-level attention and effectively improving performance.

## 2  AN EMPIRICAL ANALYSIS OF ATTENTION DILUTION

We first conduct an empirical study to analyze the attention dilution phenomenon in large language models (LLMs) during code generation. To improve the generalizability of our findings, we experimented with two different methods to compute the attention scores over input tokens. First, we used a self-attention-based method (Zhang et al., 2022; Galassi et al., 2021) to obtain self-attention scores from the last layer in LLMs, which has been shown to represent the most accurate attention distribution (Kou et al., 2024; Wan et al., 2022a). Second, we used a gradient-based method (Selvaraju et al., 2016; Shrikumar et al., 2017) that treats the entire LLM as a function and measures to what extent each input token contributes to the output. Based on these two methods, we calculate the percentage of attention on the initial prompt. Calculation details are provided in Appendix A.1.

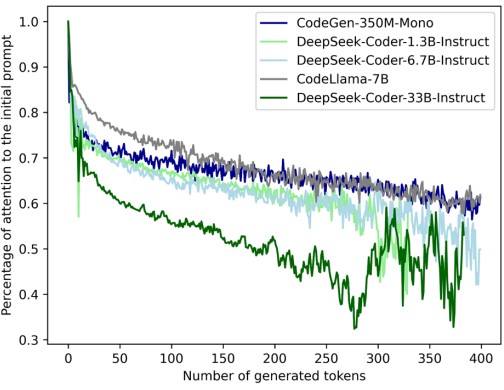 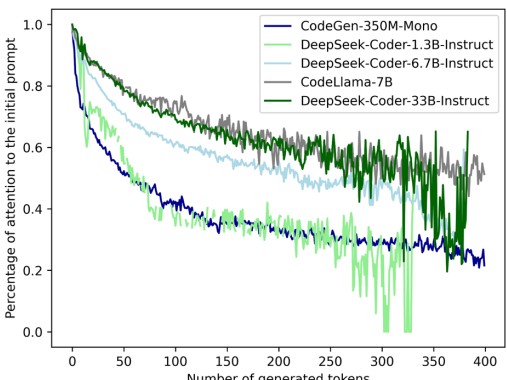

Figure 2: Shift of LLMs' self-attention to the initial prompt. The attention is calculated from the last self-attention layer of the LLM.

Figure 3: Shift of LLMs' gradient-based attention to the initial prompt. The gradient is calculated with respect to the output logits.

On HumanEval (et al., 2021c), a widely-used benchmark for code generation, we experimented with five LLMs: CodeGen-Mono-350M (Nijkamp et al., 2023), CodeLlama-7B (Rozière et al., 2024), and DeepSeek-Coder-Instruct-1.3B, 6.7B, and 33B (Guo et al., 2024). Figure 2 and Figure 3 show the evolution of the density of LLMs' attention on the initial prompt when generating the first 400 tokens. [1] The results demonstrate that as the model generates more tokens, model attention on the initial prompt gradually becomes smaller, which we refer to as *attention dilution*. Consequently, as the generated code sequence becomes longer, the code generation process becomes increasingly influenced by tokens generated in recent time steps, rather than the prompt from users. This can be problematic in two ways. First, generation errors in the previous time steps are very likely to propagate to the following steps as the model pays more attention to the preceding code tokens. Second, for complex tasks that require the generation of a long code sequence (e.g., multiple if statements), the model is likely to miss critical descriptions as it pays little attention to the user prompt deep in the code generation process.

## 3  APPROACH

### 3.1  AUTOREGRESSIVE DECODING AND ITS LIMITATIONS

Given an LLM $f_\theta$ and a prompt $x$, the model generates tokens $t_1, t_2, \ldots, t_{i-1}$ in an autoregressive manner. At step $i$, the input to $f_\theta$ is an $n \times m$ embedding matrix $\mathbf{E}_i$, defined as:

$$\mathbf{E}_i = \text{embedding}(x, t_1, t_2, \ldots, t_{i-1}) = [\mathbf{E}^x, e_1, e_2, \ldots, e_{i-1}, \texttt{PAD}]. \quad (1)$$

where $\mathbf{E}^x$ is the submatrix of embeddings for tokens in prompt $x$, $e_1, \ldots, e_{i-1}$ are embeddings of generated tokens, and $\texttt{PAD}$ is a padding submatrix.

---

[1] The average generated token number is 132. The gradual noisy plot results from a lack of lengthy generations.

The model outputs logits and transforms them into a probability distribution. Then a sampling method (e.g., greedy sampling) is applied to select the next token $t_i$:

$$t_i = \arg\max_t \text{softmax}(f_\theta(\mathbf{E}_i)) = \arg\max_t P_\theta(t|x, t_1, \ldots, t_{i-1}) \tag{2}$$

However, autoregressive decoding assumes all prior tokens are correct, giving them equal opportunity to compete for the model's attention—even when self-generated tokens may be wrong. As the number of self-generated tokens increases, the model's attention to the initial prompt (which describes the task objective) gradually dilutes. Consequently, the later a token is generated, the higher the probability that the model attends to incorrect information, increasing the likelihood of generating errors.

## 3.2 SELECTIVE PROMPT ANCHORING

To mitigate the attention dilution issue, we propose Selective Prompt Anchoring (SPA) to augment the output logits by amplifying the contextual contribution of the selective tokens within the prompt, which we refer to as "*anchored text*".

SPA introduces the mechanism of adjusting the semantic impact of selected tokens in the input matrix $\mathbf{E}_i$ towards the output logits $f_\theta(\mathbf{E}_i)$. For simplicity, here we make the entire initial prompt $x$ as the anchored text.

$\mathbf{E}_i$ is an $n \times m$ input embedding matrix at step $i$, and $\mathbf{E}^x$ represents a $n \times k$ submatrix within $\mathbf{E}_i$ covering the first $k$ columns (corresponding to the prompt $x$). They are visualized below:

$$\mathbf{E}_i = \left[ \begin{array}{ccccccc} e_{11} & \cdots & e_{1k} & e_{1,k+1} & \cdots & e_{1m} \\ e_{21} & \cdots & e_{2k} & e_{2,k+1} & \cdots & e_{2m} \\ \vdots & \ddots & \vdots & \vdots & \ddots & \vdots \\ e_{n1} & \cdots & \underbrace{e_{nk}}_{\mathbf{E}^x} & e_{n,k+1} & \cdots & e_{nm} \end{array} \right]. \tag{3}$$

We construct two $n \times m$ matrices, $\mathbf{X}$ and $\mathbf{G}_i$, which add up to $\mathbf{E}_i$. Matrix $\mathbf{X}$ is created by preserving the first $k$ columns of $\mathbf{E}_i$ corresponding to $\mathbf{E}^x$ and setting all other columns to zero (note that $\mathbf{E}^x$ and $\mathbf{X}$ remain unchanged during generating new tokens). Matrix $\mathbf{G}_i$ is constructed by setting the first $k$ columns of $\mathbf{E}_i$ that correspond to $\mathbf{E}^x$ to zero, and retaining all other elements from the remaining columns. They are visualized as follows:

$$\mathbf{X} = \begin{bmatrix} e_{11} & e_{12} & \cdots & e_{1k} & 0 & \cdots & 0 \\ e_{21} & e_{22} & \cdots & e_{2k} & 0 & \cdots & 0 \\ \vdots & \vdots & \ddots & \vdots & \vdots & \ddots & \vdots \\ e_{n1} & e_{n2} & \cdots & e_{nk} & 0 & \cdots & 0 \end{bmatrix}, \mathbf{G}_i = \begin{bmatrix} 0 & 0 & \cdots & 0 & e_{1,k+1} & \cdots & e_{1m} \\ 0 & 0 & \cdots & 0 & e_{2,k+1} & \cdots & e_{2m} \\ \vdots & \vdots & \ddots & \vdots & \vdots & \ddots & \vdots \\ 0 & 0 & \cdots & 0 & e_{n,k+1} & \cdots & e_{nm} \end{bmatrix}. \tag{4}$$

The sum of $\mathbf{X}$ and $\mathbf{G}_i$ reconstructs the original matrix $\mathbf{E}_i$:

$$\mathbf{E}_i = \mathbf{X} + \mathbf{G}_i. \tag{5}$$

Suppose we want to amplify the semantic impact of the submatrix $\mathbf{X}$ by a value $\omega$. $\omega > 1$ indicates semantic amplification, while $\omega < 1$ indicates semantic diminishment.

Here we need to define a semantic adjustment function $\Phi(\mathbf{X}, \omega)$ that scales the influence of $\mathbf{X}$ by $\omega$ times. Note that the original embedding matrix $\mathbf{E}_i$ corresponds to when $\omega$ equals 1:

$$\mathbf{E}_i = \Phi(\mathbf{X}, 1) + \mathbf{G}_i. \tag{6}$$

To amplify the semantic impact of the anchored prompt $x$ in the final logits, it is essentially calculating the integral of the partial derivative of $f_\theta$ with respect to $\omega$ from 0 to $\omega$[2]. Let $F_{\theta,i,x}(\omega)$ represent the

---

[2] $f_\theta$ is differentiable for backpropagation

augmented logits calculated by model $f_\theta$ at step $i$, where the impact of anchored text $x$ is scaled by $\omega$. Formally,

$$F_{\theta,i,x}(\omega) = f_\theta(\Phi(\mathbf{X}, \omega) + \mathbf{G}_i) \tag{7}$$

$$= F_{\theta,i,x}(0) + \int_0^\omega \frac{dF_{\theta,i,x}(t)}{dt} \, dt, \tag{8}$$

where $t$ is the variable of integration.

### 3.3 Augmented Logits by Approximation

Given the computational complexities of LLMs, directly solving $\int_0^\omega \frac{dF_{\theta,i,x}(t)}{dt} \, dt$ is impractical. Therefore, we approximate it by employing the Taylor expansion:

$$F_{\theta,i,x}(\omega) = F_{\theta,i,x}(0) + \omega \cdot F_{\theta,i,x}{}'(0) + \frac{\omega^2}{2!} F_{\theta,i,x}{}''(0) + \dots \tag{9}$$

Since LLMs are inherently non-linear, higher-order derivatives of the logits function are non-zero. We truncate the series after the first derivative to get an approximation, yielding:

$$F_{\theta,i,x}(\omega) \approx F_{\theta,i,x}(0) + \omega \cdot F_{\theta,i,x}{}'(0), \tag{10}$$

where the the integral part $\int_0^\omega \frac{dF_{\theta,i,x}(t)}{dt} \, dt$ in Equation 8 is approximated by $\omega \cdot F_{\theta,i,x}{}'(0)$.

To calculate $F_{\theta,i,x}(0)$, we mask tokens in the anchored text $x$ using masked embeddings. Each LLM provides at least one special token reserved for text masking, which almost has no semantic influence [3], e.g., <unk> for Code Llama (Rozière et al., 2024) and <pad> for DeepSeek-Coder (Guo et al., 2024). Each special token corresponds to a masked embedding. By replacing embeddings of $x$ with masked embeddings, we get a masked input matrix $\mathbf{E}_i^{mask}$. It ablates the semantic influence of the anchored text $x$ while the positional encoding is not affected. Thus, we can get

$$F_{\theta,i,x}(0) = f_\theta(\mathbf{E}_i^{mask}). \tag{11}$$

To calculate $F_{\theta,i,x}{}'(0)$, we use finite-difference methods to get an approximation. Assuming the interval of $1 - 0$ is sufficiently small for $F_{\theta,i,x}$, we get:

$$F_{\theta,i,x}{}'(0) \approx \frac{F_{\theta,i,x}(1) - F_{\theta,i,x}(0)}{1 - 0}. \tag{12}$$

Combining Equations 10, 11 and 12, we get the augmented logits by first-order approximation:

$$F_{\theta,i,x}(\omega) \approx F_{\theta,i,x}(0) + \omega \cdot (F_{\theta,i,x}(1) - F_{\theta,i,x}(0)) \tag{13}$$

$$= \omega \cdot f_\theta(\Phi(\mathbf{X}, 1) + \mathbf{G}_i) + (1 - \omega) \cdot f_\theta(\Phi(\mathbf{X}, 0) + \mathbf{G}_i) \tag{14}$$

$$= \omega \cdot f_\theta(\mathbf{E}_i) + (1 - \omega) \cdot f_\theta(\mathbf{E}_i^{mask}). \tag{15}$$

Based on the augmented logits $F_{\theta,i,x}(\omega)$ where the impact of the anchored text is adjusted by a given value $\omega$, a certain sampling algorithm is applied to select the particular token. SPA can be used to augment different existing sampling methods, such as greedy sampling, beam search, nucleus sampling (Holtzman et al., 2019), and more. We provide more discussion about approximation in Appendix A.2.

### 3.4 Tuning Anchoring Strength

The anchoring strength $\omega$ serves as a hyperparameter in SPA. Our experiments demonstrate an unimodal relationship between $\omega$ and the performance. As the anchoring strength $\omega$ increases, the performance first improves, reaching an optimum, and then declines with further increases of $\omega$. It is simple to tune this single hyperparameter through a few dozen instances. More details are discussed in Section 5.3.

---

[3] $F_{\theta,i,x}(0)$ does not mean setting the embedding vector to zeros. Instead, it means setting $\omega$ to zero, which replaces the original embedding for anchored text with the masked embedding that contains no semantic information. This masked embedding vector is non-zero.

## 3.5 SELECTION OF ANCHORED TEXT

While SPA can easily anchor the entire prompt, the initial prompt can significantly vary due to task differences. In some scenarios, the prompts can be lengthy and not all information is persistently important throughout the generation. Our goal is to identify and anchor the most informative tokens, which LLMs should persistently focus on, while excluding trivial details in the prompt. For code generation tasks, the prompt commonly comprises four possible components: (1) Natural language instruction or docstring; (2) Starting code snippet; (3) A list of test cases; (4) Few-shot examples. Intuitively, natural language instruction provides high-level guidance that LLM should continually consider. This is confirmed by our experiment in Section 5.4.

## 4 EXPERIMENTS

Our experiment aims to address four main research questions:

**RQ1** Can SPA effectively and consistently mitigate attention dilution and improve performance?
**RQ2** Can the anchoring strength tuned on one code generation setting be transferred to another?
**RQ3** How does the anchoring strength $\omega$ of SPA affect code generation performance?
**RQ4** How does the selection of anchored text affect code generation performance?

## 4.1 COMPARISON BASELINES

SPA requires access to the full logits generated by the large language models (LLMs), so we are unable to evaluate closed-source models, such as GPT-4o and Claude-3.5-Sonnet. We select five representative open-source code LLMs: CodeGen-Mono-350M (Nijkamp et al., 2023), CodeLlama-7B (Rozière et al., 2024), and DeepSeek-Coder-Instruct-1.3B, 6.7B, and 33B (Guo et al., 2024). These models have been fine-tuned for code generation tasks. Notably, the DeepSeek-Coder-Instruct models have been fine-tuned by instruction-tuning (Wei et al., 2022), while CodeGen-Mono and CodeLlama are standard text completion models. This selection aims to cover diverse SOTA code LLMs of different types and sizes.

## 4.2 BENCHMARKS

**HumanEval** (et al., 2021c). It includes 164 Python tasks designed by OpenAI developers. It was initially designed to evaluate Codex (et al., 2021a) and has since become a common benchmark for code generation.

**MBPP** (Austin et al., 2021). It includes 974 crowd-sourced Python tasks. However, due to the crowd workers' ambiguous or insufficient task descriptions, the MBPP authors created a sanitized version containing 427 tasks with clearer descriptions. We evaluate SPA on the sanitized version.

**HumanEval+** and **MBPP+**. Although HumanEval and MBPP are considered de facto standards for assessing code LLMs, a recent study (Liu et al., 2023a) found they lack sufficient test cases and precise problem descriptions. This has been demonstrated as an issue that can lead to an unreliable assessment of LLM-generated code (Liu et al., 2024b). Liu et al. (2023a) subsequently released HumanEval+ and MBPP+, which supplement HumanEval and MBPP with additional test cases and better instruction. We also evaluate SPA performance on HumanEval+ and MBPP+.

## 4.3 EVALUATION METRICS AND EXPERIMENT SETUP

**Evaluation Metric.** Following prior work (et al., 2021b; Kulal et al., 2019; et al., 2021a), we measure model performance using the Pass@$k$ metric, which measures whether any of the top k candidates can pass all the test cases. In our experiments, we calculate Pass@1 and Pass@10. For Pass@1, LLMs generate a single code snippet using greedy sampling. The task is considered successful only if this generated code passes all test cases. For Pass@10, LLMs generate top 10 most probable code snippets using beam search. The task is deemed successful if any of these candidates pass all test cases.

**Model Deployment.** We downloaded and deployed LLMs from Huggingface. To expedite evaluations, we apply 8-bit quantization (Frantar et al., 2023; Dettmers et al., 2022) to all models. Prior

studies (Li et al., 2024; Huang et al., 2024) have demonstrated that this approach has very little impact on LLM performance. All experiments were conducted on a 64-bit Ubuntu 22.04 LTS system, equipped with an AMD EPYC 7313 CPU, eight NVIDIA A5500 GPUs, and 512GB of memory. The experiments ran for approximately seven weeks.

**Prompt Design.** We use the original task descriptions from the datasets as prompts for the text-completion models, CodeLlama and CodeGen-Mono. For the three DeepSeek-Coder-Instruct models, we format the prompts using the official chat template from HuggingFace.

**Hyperparameter Tuning.** For each model and dataset, we use grid search to tune the anchoring strength $\omega$ on 1/5 tasks in dataset to get $\text{SPA}_{tuned}$. We also get the optimal anchoring strength by tuning on the entire dataset ($\text{SPA}_{optimal}$). We evaluate both of them on the remaining 4/5 dataset and compute Pass@1 via greedy search as well as Pass@10 via beam search (elaborated in Appendix A.6).

## 5 RESULTS

### 5.1 MODEL PERFORMANCE IMPROVEMENTS

Table 1: Pass@1 and Pass@10 (%) with and without using SPA

| Model | Size | HumanEval | | HumanEval+ | | MBPP | | MBPP+ | |
|---|---|---|---|---|---|---|---|---|---|
| | | Pass@1 | Pass@10 | Pass@1 | Pass@10 | Pass@1 | Pass@10 | Pass@1 | Pass@10 |
| **CodeGen-Mono** | (350M) | 15.3 | 36.6 | 12.2 | 33.6 | 19.6 | 47.7 | 15.9 | 42.4 |
| + $\text{SPA}_{tuned}$ | | 18.3 (+3.0) | 38.2 (+1.6) | 16.0 (+3.8) | 36.6 (+3.0) | 24.9 (+5.3) | 52.6 (+4.9) | 20.6 (+4.7) | 42.1 (-0.3) |
| + $\text{SPA}_{optimal}$ | | 18.3 (+3.0) | 38.2 (+1.6) | 16.0 (+3.8) | 36.6 (+3.0) | 24.9 (+5.3) | 52.6 (+4.9) | 20.6 (+4.7) | 42.1 (-0.3) |
| **DeepSeek-Coder** | (1.3B) | 66.4 | 73.3 | 61.8 | 68.7 | 58.2 | 67.0 | 52.4 | 63.7 |
| + $\text{SPA}_{tuned}$ | | 69.5 (+3.1) | 73.3 (+0.0) | 66.4 (+4.6) | 69.0 (+0.3) | 59.1 (+0.9) | 68.4 (+1.4) | 52.4 (+0.0) | 64.3 (+0.6) |
| + $\text{SPA}_{optimal}$ | | 71.0 (+4.6) | 73.3 (+0.0) | 66.4 (+4.6) | 69.5 (+0.8) | 61.7 (+3.5) | 69.3 (+2.3) | 53.4 (+1.0) | 64.3 (+0.6) |
| **DeepSeek-Coder** | (6.7B) | 75.6 | 84.0 | 70.2 | 77.9 | 67.0 | 79.8 | 58.5 | 70.2 |
| + $\text{SPA}_{tuned}$ | | 83.2 (+7.6) | 85.5 (+1.5) | 75.6 (+5.4) | 80.9 (+3.0) | 69.6 (+2.6) | 84.5 (+4.7) | 60.2 (+1.7) | 72.5 (+2.3) |
| + $\text{SPA}_{optimal}$ | | 84.0 (+8.4) | 85.5 (+1.5) | 76.3 (+6.1) | 81.7 (+3.8) | 72.2 (+5.2) | 83.6 (+3.8) | 61.1 (+2.6) | 73.4 (+3.2) |
| **CodeLlama** | (7B) | 33.6 | 58.0 | 28.2 | 48.9 | 50.9 | 61.0 | 40.8 | 49.0 |
| + $\text{SPA}_{tuned}$ | | 40.5 (+6.9) | 62.6 (+4.6) | 33.6 (+5.4) | 52.7 (+3.8) | 52.9 (+2.0) | 63.7 (+2.7) | 43.1 (+2.3) | 50.9 (+1.9) |
| + $\text{SPA}_{optimal}$ | | 41.2 (+7.6) | 64.9 (+6.9) | 35.9 (+7.7) | 54.2 (+5.3) | 52.9 (+2.0) | 63.7 (+2.7) | 43.1 (+2.3) | 51.7 (+2.7) |
| **DeepSeek-Coder** | (33B) | 81.7 | 88.5 | 77.1 | 80.2 | 73.4 | 86.8 | 63.2 | 75.8 |
| + $\text{SPA}_{tuned}$ | | 84.7 (+3.0) | 89.3 (+0.8) | 77.9 (+0.8) | 81.7 (+1.5) | 77.2 (+3.8) | 88.6 (+1.8) | 68.5 (+5.3) | 74.9 (-0.9) |
| + $\text{SPA}_{optimal}$ | | 85.5 (+3.8) | 89.3 (+0.8) | 78.6 (+1.5) | 80.9 (+0.7) | 77.2 (+3.8) | 88.0 (+1.2) | 68.5 (+5.3) | 77.2 (+1.4) |

The results in Table 1 show that SPA consistently improves Pass@1 and Pass@10 across all benchmarks and LLMs (**RQ1**). The improvement reaches up to 7.6% on HumanEval for DeepSeek-Coder (6.7B). Remarkably, through selective text anchoring, the smaller version of DeepSeek-Coder (6.7B) outperforms its much larger counterpart (33B). While Pass@10 improvements are less pronounced than Pass@1, they still demonstrate consistent enhancements across most settings. One potential reason is that SPA not only increases the accuracy of top logits but also amplifies noises in lower-ranked logits. We discuss this in detail in Appendix A.6. To better demonstrate how SPA effectively anchors LLM's attention on the initial prompt, we include two code generation examples in Appendix A.4.

Note that the performance improvement is achieved only by amplifying the original prompt's influence without introducing new knowledge or fine-tuning model parameters. We attribute SPA's effectiveness to two reasons. First, when generating a new token, each prior token carries a risk of being incorrectly attended to by the model. As the model generates more tokens that compete for attention, the likelihood of attending to irrelevant tokens increases, thereby leading to errors. In contrast, the original prompt represents the high-level user intent that is persistently relevant to generated tokens. Anchoring the model's attention on the original prompt via SPA essentially enlarges the reliable portion of the model's attention, thereby generating more accurate next tokens. Second, while each self-generated token carries a probability to be error, autoregressive decoding assumes all prior tokens are correct. This allows for error propagation as more tokens are generated. By downplaying

self-generated tokens, SPA essentially provides a fairer attention distribution by measuring the trustworthiness of prior tokens. We discuss more in Appendix A.7.

## 5.2 CROSS-DATASET & CROSS-MODEL EVALUATION

SPA introduces a single hyperparameter, anchoring strength $\omega$, which modulates the degree of the anchoring effect of SPA. We investigate the transferability of this hyperparameter across different models and datasets (**RQ2**). Firstly, we conduct a *cross-dataset* evaluation between HumanEval/HumanEval+ and MBPP/MBPP+, which have distinct prompt formats. We tune $\omega$ on HumanEval+ and evaluate Pass@1 on MBPP and MBPP+, and vice versa[4] (denoted as $\text{SPA}_{cross-dataset}$). We calculate average Pass@1 improvements on original and plus versions across all baseline models. Secondly, we perform a *cross-model* evaluation by tuning $\omega$ on one model and evaluating Pass@1 on the remaining four. For each model, we compute the average Pass@1 improvements across all the other models, for HumanEval/HumanEval+ and MBPP/MBPP+ respectively (denoted as $\text{SPA}_{cross-model}$). Similar to Section 5, $\text{SPA}_{tuned}$ represents tuning within the split partial dataset, while $\text{SPA}_{optimal}$ represents tuning within the entire dataset.

Table 2: Pass@1 improvements (%) based on cross-dataset tuning

| Dataset | $\text{SPA}_{cross-dataset}$ | $\text{SPA}_{cross-model}$ | $\text{SPA}_{tuned}$ | $\text{SPA}_{optimal}$ |
|---|---|---|---|---|
| HumanEval/+ | + 2.01 | - 0.29 | + 4.36 | + 5.11 |
| MBPP/+ | + 2.50 | + 0.37 | + 2.86 | + 3.57 |

As shown in Table 2, we find the anchoring strength $\omega$ tuned on one model is hardly transferred to another. However, $\omega$ tuned on one dataset can be transferred to another with reduced but still effective performance. These observations suggest that the anchoring strength is highly model-dependent and partially task-dependent.

## 5.3 ANALYSIS OF ANCHORING STRENGTH

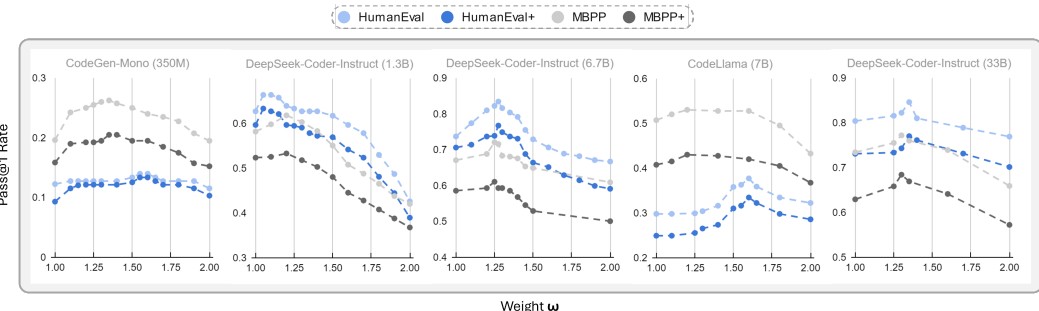

Figure 4: Analysis of Anchoring Strength

To further investigate the relationship between code generation performance and the anchoring strength of SPA (**RQ3**), Figure 4 illustrates the change in Pass@1 for various values of $\omega$ across each model and benchmark ($\omega = 1$ represents the original model). We observe a roughly unimodal relationship between $\omega$ and performance: as $\omega$ increases, performance first improves, reaches an optimum, and then declines with further increases. While the optimal $\omega$ varies slightly across different models and benchmarks, it tends to be model-dependent. Furthermore, we find that any $\omega$ value below 1.25 leads to performance improvements across all scenarios.

## 5.4 ANALYSIS OF ANCHORED TEST SELECTION

To investigate the impact of anchored text selection in code generation tasks (**RQ 4**), we calculate pass@1 by masking different components in the prompt. Prompts in HumanEval/HumanEval+

---

[4] The "plus" versions of HumanEval and MBPP share identical prompts with their original counterparts, so we can only tune on the plus version.

include the function signature (referred to as **Code**), natural language task descriptions (**NL**), and test cases (**Test**). Prompts in MBPP/MBPP+ consist of task descriptions (**NL**) followed by test cases (**Test**). For HumanEval/HumanEval+, we create four conditions by removing test cases and source code. For MBPP and MBPP+, we create two conditions by removing test cases. We chose to anchor the natural language task descriptions in all conditions, as they serve as the core task intent to mitigate attention dilution. We use SPA tuned on the entire dataset in all conditions. For each condition and benchmark, we calculate the average Pass@1 improvement across all five models.

Table 3: Improvements of Pass@1 rates (values in %) for different anchored text

| Anchored Text | HumanEval | HumanEval+ | MBPP | MBPP+ |
|---|---|---|---|---|
| *NL* | **+ 5.48** | **+ 5.08** | **+ 4.26** | **+ 3.22** |
| *NL + Test* | + 5.11 | + 4.89 | + 4.05 | + 3.11 |
| *NL + Code* | + 4.87 | + 4.65 | – | – |
| *NL + Code + Test* | + 4.76 | + 4.57 | – | – |

Table 3 shows that anchoring the natural language task description alone yields the best performance. This implies that anchoring more tokens in the prompt is not necessarily beneficial. Anchoring an increasing number of tokens can diminish the effectiveness of differentiating the logit distribution. For example, anchoring all tokens would merely introduce random noise. Instead, focusing on fewer but critical, informative tokens leads to better results. More specifically, the optimal anchored tokens should be those highly relevant to the current context but overlooked by the model.

In code generation tasks, the natural language task description represents the user's intent, which is persistently relevant. Continuously anchoring this part provides a sub-optimal but effective trade-off solution. While opportunities exist to further refine the range of critical tokens by filtering out less relevant ones, we find this requires significant effort in studying and designing such an algorithm. For other tasks, the range of anchored text can vary significantly. For instance, unlike natural language task descriptions in code generation tasks, code translation tasks lack a component that needs persistent anchoring. Additionally, the anchored text may also vary across different models—some tokens may be easily overlooked by certain models but correctly attended to by others.

## 6 RELATED WORK

**Code Generation.** In recent years, there has been rapid progress in the development of code generation approaches (Dong & Lapata, 2016; Iyer et al., 2018) and benchmarks (et al., 2021c; Austin et al., 2021; Liu et al., 2023a; Hendrycks et al., 2021). With the advent of large language models (LLMs), such as GPT-4 (OpenAI & et al., 2024) and Gemini (Team & et al., 2024), code generation has become a standard capability. Subsequent research has focused on fine-tuning these pre-trained LLMs to achieve state-of-the-art performance.

Despite their remarkable ability to follow natural language instructions, LLMs still face challenges when generating long and complex code. To enhance the code generation capabilities of LLMs, recent studies have explored train-free approaches such as prompt engineering (Denny et al., 2023; White et al., 2023), in-context learning (Dong et al., 2023; Li et al., 2023a;b), and retrieval-augmented generation (Lewis et al., 2020; Du et al., 2024). Additionally, self-debugging techniques (Chen et al., 2023) enable LLMs to debug code based on error messages and execution results, while self-planning (Jiang et al., 2023) allows LLMs to decompose tasks into subtasks and implement solutions step-by-step. The chain-of-thought approach (Le et al., 2024; Suzgun et al., 2022; Ma et al., 2023) facilitates a step-by-step reasoning process in LLMs. Complementing these approaches, SPA introduces an orthogonal approach particularly suitable for code generation. It can be integrated with existing methods to further improve performance.

**Controllable Generation.** Compared to fine-tuning a language model (LM) at the decoding time, controllable generation aims to steer the pre-trained LMs to match a sentence-level attribute (e.g., a topic on sports). Existing approaches usually require additional models or training, such as fine-tuning a smaller LM (Liu et al., 2024a; 2021; Yang & Klein, 2021; Dathathri et al., 2020), a reward model (Deng & Raffel, 2023; Lu et al., 2023), or a fine-tuned model with controlling codes (Krause et al., 2021; Li & Liang, 2021; Keskar et al., 2019). The mechanism used in SPA can also be used

to control the generation by adjusting anchoring strength over the input text. Compared to the aforementioned works, SPA does not require any additional models or training.

**Logit Arithmetic.** There has been a growing body of methods that perform arithmetic on multiple logit distributions to enhance text generation. These methods include contrasting logits from multiple LMs (Liu et al., 2024a; 2021; Dou et al., 2019; Zhao et al., 2024), logits of LMs of different sizes (Li et al., 2023d), logits from different layers of a model (Chuang et al., 2024; Gera et al., 2023), and logits from the same model given different inputs (Pei et al., 2023; Shi et al., 2023; Malkin et al., 2022; Sennrich et al., 2024; Leng et al., 2023). Similar ideas have also been explored in diffusion models (Han et al., 2024; Ho & Salimans, 2022).

SPA can be considered analogous to contrasting logits from the same model when given different inputs. However, we delve deeper by modeling a mathematical approximation of semantic adjustment over arbitrary groups of embeddings. Furthermore, SPA is specifically designed to address the attention dilution issue in LLMs during code generation—a phenomenon first observed in our work. By contrast, none of existing works explored code generation tasks. They primarily focus on reducing hallucinations (Shi et al., 2023; Sennrich et al., 2024; Leng et al., 2023), enhancing coherence (Malkin et al., 2022), factuality (Chuang et al., 2024), and controllable text generation (Liu et al., 2021; Pei et al., 2023; Zhao et al., 2024). Besides, SPA focuses on perturbation of the original prompt through masking rather than providing additional context (Pei et al., 2023; Shi et al., 2023; Malkin et al., 2022) or changing to a completely new prompt (Sennrich et al., 2024).

# 7 LIMITATIONS & FUTURE WORK

We employed 8-bit quantized LLMs to expedite all experiments. Although this method has been shown to have minimal impact on performance, we did notice some degradation. Furthermore, we did not evaluate very large LLMs (e.g., more than 100B) due to computational constraints. Despite the unimodal feature, it is infeasible to enumerate all the anchoring strength $\omega$ on the continuous distribution. The real optimal $\omega$ should perform slightly better than the values reported in Section 4.

While SPA achieved a consistent improvement on LLMs with different sizes and types (i.e., instruction-tuned & text completion), we do not observe a monotonic relationship between model attributes and the improvement. Furthermore, there is no obvious correlation between the original model performance and the improvement. It is an interesting future direction to investigate how different model attributes affect the improvement achieved by SPA. Given the performance improvements, the computational overhead of SPA is acceptable. We elaborate on this in Appendix A.5.

The effectiveness of SPA highlights its potential in other domains, particularly for generation tasks. However, we believe rigorous experiments are necessary to confirm whether attention dilution exists in other tasks, as different tasks may have unique input and output patterns. Investigating the existence of attention dilution and determining which text to anchor in other tasks presents an interesting avenue for future research. In this work, we pre-define the method for selecting anchored tokens and use a fixed anchoring strength when generating code. We consider this approach a baseline. Future work could explore dynamically determining both the anchored text and the anchoring strength based on different contexts and sampling stages. Furthermore, the underlying principle of SPA is not confined to transformer-based LLMs and could be adapted for use in other model architectures (e.g., RNNs).

# 8 CONCLUSION

In this paper, we propose SPA, a model-agnostic approach designed to enhance the quality of code generated by large language models (LLMs) by mitigating the attention dilution issue. SPA employs a novel technique to adjust the influence of selected groups of input tokens, based on a mathematical approximation. Our empirical study indicates that LLMs may overlook the initial prompt as generating more new tokens. By amplifying the initial prompt's influence throughout code generation, SPA consistently and significantly improves performance across models of various sizes on multiple benchmarks.

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

## A APPENDIX / SUPPLEMENTAL MATERIAL

### A.1 ATTENTION CALCULATION

**Self-attention.** Most LLMs are based on the decoder of transformer (Vaswani et al., 2017) which has multiple self-attention layers. Roughly speaking, given an LLM $f_\theta$ and an input sequence of tokens

$t_0, t_1, \ldots, t_n$ where $t_i$ represents the $i$th token. The transformer calculates relevance scores between every pair of tokens. The self-attention score for a token $t_i$ in the sequence can be roughly formulated as:

$$\text{attention}(t_i) \approx \frac{\sum_{j=1}^{n} \text{relevance}(t_i, t_j)}{\sum_{i=1}^{n} \sum_{j=1}^{n} \text{relevance}(t_i, t_j)}, \qquad (16)$$

where the relevance function approximates the computation among $Q, K, V$ in transformers (Vaswani et al., 2017). However, different layers have different attention distributions. According to a study (Wan et al., 2022b), deeper self-attention layers can better capture long-distance dependencies and program structure, so we calculate the attention by aggregating attention from multiple heads at the last layer. Nevertheless, this still excludes the influence from the last forward layer.

**Gradient-based Attention.** Compared to using self-attention layers in transformers, the gradient-based method can be generalized to different model architectures and consider the entire model as a whole. It computes the model's attention by calculating the gradients relative to the input. Intuitively, a token that induces a larger gradient is considered more influential, suggesting that the model pays greater attention to it. Formally, the attention over the token $t_i$ is calculated by

$$\text{attention}(t_i) = \frac{\partial f_\theta(t_0, t_1, \ldots, t_n)}{\partial t_i}. \qquad (17)$$

**Attention Percentage to the Prompt.** Based on these two methods, we analyze how the attention of LLMs to the initial prompt shifts. Formally, given the prompt $x$ and the following generated tokens $t_0, t_1, \ldots, t_{i-1}$, we calculate the percentage of attention $\alpha(x)$ over the initial prompt

$$\alpha(x) = \frac{\text{attention}(x)}{\text{attention}(x) + \sum_{i=1}^{n} \text{attention}(t_i)} \qquad (18)$$

Given attention analysis requires open sourcing, we select five SOTA code LLMs with various sizes. We run the experiments on HumanEval (et al., 2021c), one of the most popular benchmarks for evaluating code generation models. We run five LLMs (Nijkamp et al., 2023; Rozière et al., 2024; Guo et al., 2024) on all 164 Humaneval tasks. Figure 2 shows the self-attention shift and Figure 3 shows the gradient-based attention shift when generating the first 400 tokens. The value gradually becomes noisy due to the lack of generated sequence with enough length.

The results demonstrate that there indeed exists such attention dilution issue. Due to the autoregressive nature, LLMs' attention to the initial prompt is gradually diluted as generating more code. LLMs tend to attend to code generated by itself. Our finding is supported by another study (Chiang & Cholak, 2022) which investigates the self-attention dilution of transformers in a more general scenario.

A.2 APPROXIMATION IN SPA

In Equation 10, we get the approximation by only keeping the first derivative in Equation 9, but it is also feasible to calculate a higher-order approximation. For example, if we want to keep the term involving the second-order derivative $\frac{\omega^2}{2!} F_{\theta,i,x}''(0)$, it can still be computed using finite-difference methods:

$$F_{\theta,i,x}''(0) \approx \frac{F_{\theta,i,x}(1) - 2F_{\theta,i,x}(0) + F_{\theta,i,x}(-1)}{(1-0)^2}. \qquad (19)$$

$F_{\theta,i,x}(-1)$ can be solved by Equation 13 where $F_{\theta,i,x}(0)$ and $F_{\theta,i,x}(1)$ are the logits generated from the original input and the logits generated from the masked input.

However, no matter how many terms we keep in Equation 9, we find we can only represent $F_{\theta,i,x}(\omega)$ as a linear combination of $F(0)$ and $F(1)$, weighted by an unknown variable $\omega$.

In Section 5.3, our experiments reveal that $\omega$'s impact on code generation performance follows an unimodal pattern—initially increasing, then decreasing. Due to its distribution simplicity, we argue that while a higher-order approximation may yield a more reasonable performance distribution across different $\omega$ values, it does not significantly affect the process of locating the optimal anchoring strength. Therefore, beyond its computational efficiency, the first-order approximation in SPA is adequate for calculating semantically accurate augmented logits.

Table 4: Optimal $\omega$ for each model and benchmark

| Model | HumanEval | HumanEval+ | MBPP | MBPP+ | Average |
|---|---|---|---|---|---|
| **CodeGen-Mono** (350M) | 1.20 | 1.20 | 1.35 | 1.35 | 1.28 |
| **DeepSeek-Coder** (1.3B) | 1.05 | 1.05 | 1.20 | 1.20 | 1.13 |
| **DeepSeek-Coder** (6.7B) | 1.28 | 1.28 | 1.25 | 1.25 | 1.26 |
| **CodeLlama** (7B) | 1.60 | 1.60 | 1.20 | 1.20 | 1.40 |
| **DeepSeek-Coder** (33B) | 1.35 | 1.35 | 1.30 | 1.30 | 1.33 |
| Average | 1.30 | 1.30 | 1.33 | 1.33 | 1.28 |

### A.3 OPTIMAL ANCHORING STRENGTH

Table 4 reports optimal anchoring strength values $\omega$ that are used in our main results (Table 1). We observe the average value of 1.28 can be used to effectively improve performance across all benchmarks for all LLMs.

### A.4 EXAMPLES

Figure 5 presents two examples comparing the code generated by models alone and the models augmented using SPA.

In the first example, CodeLlama (7B) overlooks the specified condition "upper vowels." In contrast, SPA enhances the model's focus on the intended purpose. The code initializes all the upper vowels in the first line and correctly refers to it later.

In the second example, DeepSeek-Coder (1.3B) erroneously sorts the list by string names instead of integers. When using SPA, the model demonstrates improved recognition of the required procedures, aligning more closely with the task description. The code correctly sorts and reverses the list. Then the integer list is mapped to the string list.

### A.5 COMPUTATIONAL COST

In our implementation, SPA requires twice the inference time to obtain two logits, plus some minor additional computation costs for operations like logit addition. We observe that SPA typically takes 2 to 3.5 times longer than regular inference. There is little extra memory overhead. Compared to the size of the LLM, SPA only requires a few additional variables and an embedding matrix to buffer in the RAM.

We believe our implementation can be further optimized for speed. For example, there is a significant overlap between the masked embedding and the original embedding. This overlap can be leveraged for acceleration through caching repetitive computations in transformer Liu et al. (2023b); Ge et al. (2024).

### A.6 BEAM SEARCH WITH SPA

To calculate Pass@10 in Section 5, we employ beam search to generate 10 candidate code snippets. When running beam search with SPA, however, we found that directly sampling top beams based on the augmented logits produced by SPA led to performance degradation.

We hypothesize that this phenomenon occurs because while SPA successfully amplifies the influence of anchored text and improves the accuracy of top logits, it also amplifies noise in lower-ranked logits. This undermines the reliability of the overall probability distribution, thereby hindering the sampling process.

① 
```python
def count_upper(s):
    """
    Given a string s, count the number
    of uppercase vowels in even indices.

    For example:
    count_upper('aBCdEf') returns 1
    count_upper('abcdefg') returns 0
    count_upper('dBBE') returns 0
    """
    count = 0
    for index, letter in enumerate(s):
        if index % 2 == 0:
            count += 1
    return count
```
✗

Selective Anchored Text
```python
def count_upper(s):
    """
    Given a string s, count the number
    of uppercase vowels in even indices.

    For example:
    count_upper('aBCdEf') returns 1
    count_upper('abcdefg') returns 0
    count_upper('dBBE') returns 0
    """
    vowels = 'AEIOU'
    count = 0
    for i in range(0, len(s), 2):
        if s[i].upper() in vowels:
            count += 1
    return count
```
✓

② 
```python
def by_length(arr):
    """
    Given an array of integers, sort the integers
    that are between 1 and 9 inclusive, reverse the
    resulting array, and then replace each digit by
    its corresponding name from
    "One", "Two", "Three",
    "Four", "Five", "Six",
    "Seven", "Eight", "Nine".

    For example:
    by_length([2, 1, 8, 2, 3])
      returns ["Eight", "Three", "Two", "Two", "One"]
    by_length([])
      returns []
    by_length([1, -1 , 55])
      returns ['One']
    """

    num_to_word= {
    1: 'One', 2: 'Two', 3: 'Three',
    4: 'Four', 5: 'Five', 6: 'Six',
    7: 'Seven', 8: 'Eight', 9: 'Nine'}

    result = []

    for num in arr:
        if 1 <= num <= 9:
            result.append(num_to_word[num])
    print(result)
    result.sort(reverse=True)
    return result
```
✗

Selective Anchored Text
```python
def by_length(arr):
    """
    Given an array of integers, sort the integers
    that are between 1 and 9 inclusive, reverse the
    resulting array, and then replace each digit by
    its corresponding name from
    "One", "Two", "Three",
    "Four", "Five", "Six",
    "Seven", "Eight", "Nine".

    For example:
    by_length([2, 1, 8, 2, 3])
      returns ["Eight", "Three", "Two", "Two", "One"]
    by_length([])
      returns []
    by_length([1, -1 , 55])
      returns ['One']
    """

    num_to_name = {
    1: 'One', 2: 'Two', 3: 'Three',
    4: 'Four', 5: 'Five', 6: 'Six',
    7: 'Seven', 8: 'Eight', 9: 'Nine'}

    valid_nums = [num for num in arr if 1 <= num <= 9]
    valid_nums.sort()
    valid_nums.reverse()
    names = [num_to_name[num] for num in valid_nums]
    return names
```
✓

Figure 5: Examples of generated code by LLMs alone (left) and using SPA (right).

To address this issue, we retrieve top candidate tokens based on the augmented logits but use original probabilities to compute beam probability. This ensures that important, potentially overlooked tokens are considered while maintaining reliable probabilities.

As demonstrated in our experiment results (Table 1), the improvements in Pass@10 are less effective than Pass@1. We posit that fully leveraging the power of SPA requires a more sophisticated beam search algorithm specifically adapted to SPA. We leave this as an avenue for future work.

### A.7 HYPOTHESIZED EXPLANATION FOR ATTENTION DILUTION AND SPA'S EFFECTIVENESS

SPA is motivated by a recent study Kou et al. (2024) and our empirical observations demonstrating the attention dilution issue. Our experiment results in Section 5 echo our observation and confirm the existence of attention dilution during code generation. Here we propose a detailed explanation for this phenomenon based on our knowledge and hypotheses. We believe it stems from two limitations in regular decoding: (1) **Distraction** and (2) **Error propagation**.

**Distraction.** When a transformer generates a token, its correctness depends on two abilities: (1) whether the model attends to the correct context, and (2) whether the model can derive the correct token based on this context. SPA aims to improve the first ability. Suppose we have a perfect transformer. For each generated token, it should only attend to relevant prior tokens and ignore irrelevant ones. However, no model is perfect. For each prior token, there is a chance the model incorrectly identifies and attends to it. More tokens mean a higher probability that the attention contains an error, thereby leading to distraction.

While self-generated tokens are also important context, they are less persistently related than task description in code generation. Amplifying the task description via SPA can improve attention reliability, thereby mitigating distraction.

**Error propagation.** Compared to reliable task description tokens, the self-generated code tokens may be wrong. However, autoregressive decoding assumes all prior tokens are correct, and all the tokens have an equal opportunity to compete for the model's attention. As a result, the later a token is generated, the higher the probability it is wrong as errors propagate. SPA adds extra attention to earlier tokens that are less likely to be incorrect, creating a fairer attention distribution.

