# OpenReview forum: "Selective Prompt Anchoring for Code Generation"
_ICLR.cc/2025/Conference — Submitted to ICLR 2025_

### Official Review · Reviewer_2FhE · 2024-10-20

**Soundness:** 3
**Presentation:** 3
**Contribution:** 3
**Rating:** 6
**Confidence:** 4

**Summary:**

The authors identify that LLMs tend to dilute their self-attention on the initial prompt as more code tokens are generated, leading to inaccuracies in the generated code. To address this, they propose a novel approach called Selective Prompt Anchoring (SPA), which amplifies the influence of selected parts of the initial prompt, referred to as "anchored text," during the code generation process.

Key contributions :

**1 Identification of Attention Dilution:** The authors conduct an empirical study revealing that LLMs' attention to the initial prompt diminishes as more code is generated, which they term as "attention dilution."

**2 Proposal of Selective Prompt Anchoring (SPA):** SPA is introduced as a model-agnostic method to optimize LLMs' attention by amplifying the contextual contribution of selective prompt text towards each generated token.

**3** SPA calculates the logit distribution difference with and without the anchored text, approximating the contextual contribution of the anchored text to the output logits. It then creates an augmented logit distribution by linearly combining the original logit distribution and the logit difference.

**Strengths:**

**1 Rigorous Theoretical Foundation:** The authors provide a detailed theoretical proof for the concept of Selective Prompt Anchoring (SPA), demonstrating how it can approximate the contextual contribution of anchored text to the output logits. This theoretical underpinning adds depth to the methodology and supports the validity of the approach.

**2 Comprehensive Experimental Validation:** The paper backs the theoretical contributions with extensive experimental validation across multiple LLMs of varying sizes. The consistent performance improvements observed on different benchmarks and models showcase the robustness and generalizability of the SPA approach.

**Weaknesses:**

**1** The authors assume that self-generated tokens by the model are potentially incorrect, which leads to the idea that anchored text should only come from the initial prompt. However, it's possible that some self-generated tokens could also be considered as anchored text, which SPA does not account for currently.

**2** It's not clear how SPA compares to other state-of-the-art methods which focused on anchor-attention[1] improvements . More comprehensive benchmarking against a broader range of methods could strengthen the paper's claims.

**3** The selection of anchored text is crucial for the effectiveness of SPA, yet the paper does not provide a method for identifying the most informative tokens within the prompt. The authors conduct experiments in a general manner, but a more nuanced approach to selecting anchored text could potentially improve results.

**4** The experiments are conducted on HumanEval and MBPP datasets, which may not fully represent the complexity and diversity of real-world programming tasks. Testing SPA on more challenging datasets like OpenEval[2] or BigCodeBench[3] could provide a better understanding of its performance under more demanding conditions.

**5** The paper does not address the generalization of SPA across different programming languages. It's unclear if the same experimental conclusions would hold for languages other than those tested in the paper.

[1] [Anchor-based Large Language Models](https://aclanthology.org/2024.findings-acl.295) (Pang et al., ACL Findings 2024)

[2] [Chain-of-thought in neural code generation: From and for lightweight language models](https://ieeexplore.ieee.org/abstract/document/10634302/) (Yang et al., TSE 2024)

[3] https://bigcode-bench.github.io/

**Questions:**

**1** What are the advantages of SPA over Anchor-LLM with improved attention mechanism? (I looked at the open source code and it seems that SPA only supports greedy search at the moment)

**2** How well does SPA generalize to other programming languages besides python?

**3** Why there is no consideration on how to choose the optimal anchored-text?

---

> ### Author Response · Authors · 2024-11-18
> **Response to Reviewer 3**
>
> Thanks for your insightful comments! We reply to each question (Weakness) below.
>
> ---
>
> # Question 1 (Weakness 2)
>
> Thanks for introducing Anchor-LLM. While their names are similar, we find that SPA and Anchor-LLM are different approaches that address different challenges.
> Anchor-LLM aims to compress semantic information into anchor tokens, thus reducing the need for KV cache. They aim to decrease the time and space requirements for LLM decoding, while not compromising performance a lot. By contrast, SPA introduces a mechanism to amplify/reduce the influence of certain tokens in the prompt, thereby controlling the LLM decoding direction. We found amplifying attention over the original prompt can consistently improve code generation performance. Therefore, SPA and Anchor-LLM are not directly comparable. We think they can be integrated together to achieve both computationally efficient and more accurate decoding.
>
> ---
>
> # Question 2 (Weakness 4 & 5)
>
> Thank you for your suggestion. We have conducted additional experiments to evaluate SPA on other programming languages via HumanEval-X [1]. Below is the results. We will add the results in paper.
>
>
> | Model | Python | Java | JavaScript | C++ | Go |
> |-------|---------|------|------------|-----|-----|
> | Codegen-350M | 15.3% | 9.8% | 13.4% | 9.8% | 6.7% |
> | +SPA | 18.3% | 11.6% | 15.9% | 12.2% | 11.0% |
> | DeepSeek-Coder-1.3B | 66.4% | 42.7% | 57.3% | 43.3% | 40.2% |
> | +SPA | 69.5% | 45.1% | 59.8% | 45.1% | 42.1% |
> | DeepSeek-Coder-6.7B | 75.6% | 48.8% | 65.2% | 49.4% | 45.7% |
> | +SPA | 83.2% | 53.7% | 72.0% | 50.0% | 50.0% |
> | CodeLlama-7B | 33.6% | 22.0% | 29.3% | 22.0% | 20.1% |
> | +SPA | 40.5% | 26.2% | 34.8% | 26.2% | 24.4% |
> | DeepSeek-Coder-33B | 81.7% | 53.0% | 70.7% | 53.7% | 49.4% |
> | +SPA | 84.7% | 54.9% | 73.2% | 55.5% | 51.2% |
>
>
> [1] Qinkai Zheng, Xiao Xia, Xu Zou, Yuxiao Dong, Shan Wang, Yufei Xue, Lei Shen, Zihan Wang, Andi Wang, Yang Li, Teng Su, Zhilin Yang, and Jie Tang. 2023. CodeGeeX: A Pre-Trained Model for Code Generation with Multilingual Benchmarking on HumanEval-X. In Proceedings of the 29th ACM SIGKDD Conference on Knowledge Discovery and Data Mining (KDD '23). Association for Computing Machinery, New York, NY, USA, 5673–5684. https://doi.org/10.1145/3580305.3599790
>
> ---
>
> # Question 3 (Weakness 1 & 3)
>
> ## Weakness 1
>
> Thank you for your question. SPA is mainly designed to address the attention dilution issue with respect to the original prompt in code generation task. Our empirical study revealed that the model typically over-attends to recently generated tokens. Consequently, increasing attention to recent self-generated tokens will hinder addressing attention dilution issue.
>
> Furthermore, self-generated tokens can be incorrect. Overly attending to these tokens may lead to error propagation in subsequent steps (Appendix  A.7).
>
> ## Weakness 3
>
> As the model generates different tokens, its attention dynamically changes at each step. Precisely locating the "most informative" tokens at all steps is extremely challenging. A recent study [2] has shown that attention can be overly distributed to the first or special tokens (a phenomenon called "attention sink"). Furthermore, determining how the model distributes its attention to sub-tokens is complex. Overly micro-managing specific tokens can easily lead to poor performance. For instance, if we incorrectly steer the model's attention to the wrong words in just 5% of cases, the final generated code may be incorrect. Therefore, in our approach, we pursue a balanced strategy. SPA anchors the natural language (NL) instruction in the code generation prompt. We chose this method because, although it may slightly reduce precision, the NL instruction remains consistently relevant to all generated code tokens. Thus, negative influence of less relevant tokens in the prompt can be counteracted by most other tokens. We promise to add this discussion in paper.
>
> [2] Xiao, G., Tian, Y., Chen, B., Han, S., & Lewis, M. (2024, April 7). *Efficient streaming language models with attention sinks*. In The Twelfth International Conference on Learning Representations.

---

> > ### Comment · Reviewer_2FhE · 2024-11-20
> > **Two final questions.**
> >
> > Thanks to the author for the clarification, most of the doubts have been explained. I am now confused about only two issues:
> >
> > (1) Does SPA only support greedy search?
> >
> > (2) Testing SPA on more challenging datasets like OpenEval[2] or BigCodeBench[3]?

---

> ### Author Response · Authors · 2024-11-21
> **Response to Reviewer 4's Follow-up Questions**
>
> Thank you for your follow-up questions! We are glad to answer them.
>
>
>
> ### (1)
> SPA can support any decoding strategy, such as beam search or nucleus sampling. This is because SPA only augments logits of the last layer of the model, and it is irrelevant to the decoding strategy (Section 3.3). In our current experiment, we've used beam search to calculate pass@10 in Table 1. The results demonstrate that SPA can also improve beam search accuracy compared to the original model. We've provided a more detailed discussion of beam search in Appendix A.6. You can find our code implementation for beam search + SPA at Line 3202 in https://anonymous.4open.science/r/Selective-Prompt-Anchoring-3693/weighted_utils/weighted_text_utils.py.
>
>
> ### (2)
> We've conducted additional experiments on BigCodeBench. Please check the results below.
> While the absolute improvements aren't as big as in simple benchmarks, the relative improvements remain comparable. For example, although the absolute improvement for CodeGen-Mono-350M is 0.3%, SPA enhances its performance by 27% relative to the original 1.1% performance.
> This is because SPA only adjusts the attention of the code generation model and therefore still relies on the model's innate capability of code generation. In other words, if a model could solve a task but misses a few tokens or requirements in the prompt, SPA can help with this by adjusting the attention. If a model is very poor and doesn't possess the capability to solve a task, adjusting the model attention won't help much.
> We promise to include these new results and discussion in the paper.
>
>
> | Model | BigCodeBench | HumanEval | HumanEval+ | MBPP | MBPP+ |
> |-------|--------------|-----------|------------|------|--------|
> | CodeGen-Mono-350M | 1.1 | 15.3 | 12.2 | 19.6 | 15.9 |
> | +SPA | 1.4 (+0.3) (27%) | 18.3 (+3.0) (20%) | 16.0 (+3.8) (31%) | 24.9 (+5.3) (27%) | 20.6 (+4.7) (30%) |
> | DeepSeek-Coder-1.3B | 2.5 | 66.4 | 61.8 | 58.2 | 52.4 |
> | +SPA | 3.3 (+0.8) (32%) | 69.5 (+3.1) (5%) | 66.4 (+4.6) (7%) | 59.1 (+0.9) (2%) | 52.4 (+0.0) (0%) |
> | DeepSeek-Coder-6.7B | 12.7 | 75.6 | 70.2 | 67.0 | 58.5 |
> | +SPA | 14.2 (+1.5) (12%) | 83.2 (+7.6) (10%) | 75.6 (+5.4) (8%) | 69.6 (+2.6) (4%) | 60.2 (+1.7) (3%) |
> | CodeLlama-7B | 3.4 | 33.6 | 28.2 | 50.9 | 40.8 |
> | +SPA | 3.8 (+0.4) (12%) | 40.5 (+6.9) (21%) | 33.6 (+5.4) (19%) | 52.9 (+2.0) (4%) | 43.1 (+2.3) (6%) |
> | DeepSeek-Coder-33B | 18.9 | 81.7 | 77.1 | 73.4 | 63.2 |
> | +SPA | 20.7 (+1.8) (10%) | 84.7 (+3.0) (4%) | 77.9 (+0.8) (1%) | 77.2 (+3.8) (5%) | 68.5 (+5.3) (8%) |

---

> > ### Comment · Reviewer_2FhE · 2024-11-22
> >
> > Thanks for clarification.
> > Now rating is 6.

---

> > > ### Author Response · Authors · 2024-11-22
> > > **Thank you**
> > >
> > > Thank you very much! We really appreciate it. We will make sure to incorporate your feedback in our paper revision.

---

### Official Review · Reviewer_wsgt · 2024-10-27

**Soundness:** 2
**Presentation:** 2
**Contribution:** 2
**Rating:** 5
**Confidence:** 4

**Summary:**

This paper proposes that as the generated code tokens increase, LLMs often dilute their self-attention on the initial prompt, which is one of the fundamental reasons for inaccuracies in the code generation. To solve this issue, the paper introduces Selective Prompt Anchoring (SPA), a model-agnostic approach that amplifies the influence of selective prompts. The results demonstrate that, after tuning on a few dozen instances, SPA improves Pass@1 on new tasks by up to 7.6\%.

**Strengths:**

1. SPA, as a method that requires no training, can be applied to various models, demonstrating its broad applicability.
2. The method has been validated across multiple code generation models, with experimental results showing consistent performance improvements among various models.

**Weaknesses:**

1. This paper proposes to optimize the results by enhancing attention to anchored text. Similar methods have already been explored in natural language contexts, and it is recommended to include them in the "Related Work" section.
2. The authors should specify which specific information should be selected as anchored text in section 3.5, or provide a method for segment identification.

**Questions:**

1. It is recommended to include experimental comparisons with other LLM-based optimization approaches, rather than solely comparing with baselines.
2. The authors mention identifying and anchoring the most informative tokens in longer prompts, thereby excluding trivial information. However, I didn't see any methods related to identifying fine-grained informative tokens  in the paper.
3. The SPA method amplifies the influence of specific parts of the prompt. Will this approach change the model's behavior and compromise the initially correct output?

---

> ### Author Response · Authors · 2024-11-18
> **Response to Reviewer 3 (Part 1)**
>
> Thanks for your insightful comments! We reply to each weakness and question below.
>
> ---
>
> # Weakness 1
>
> Thanks for the reviewer’s comments. We'd like to emphasize that our approach differs significantly from existing methods. Current methods such as [1, 2] require extensive adaption to existing models to manipulate the model attention.
> For example, [1] requires an model profiling stage to identify attention headers to be adjusted for improvement. During inference, it recalculates attention distribution for each layer and selected headers.
> In contrast, SPA only requires tuning 1 hyper-parameter for optimal performance. During inference, SPA can adjust attention by simply computing logits difference.
> Furthermore, [1] requires user input to steer model attention, while SPA can automatically improve code generation performance by amplifying the influence of original prompt.
> [2] tunes a feature selection module to redirect the attention to task-relevant features.
> In contrast, SPA is model-agnostic and applicable to different model architectures. We promise to improve the related work section by including a more comprehensive comparison between SPA and existing approaches.
>
> [1] Zhang, Q., Singh, C., Liu, L., Liu, X., Yu, B., Gao, J., & Zhao, T. Tell Your Model Where to Attend: Post-hoc Attention Steering for LLMs. In The Twelfth International Conference on Learning Representations.
>
> [2] Shi, B., Gai, S., Darrell, T., & Wang, X. (2023, July 11). Toast: Transfer learning via attention steering. arXiv.org. https://arxiv.org/abs/2305.15542
>
>
>
> # Question 1
>
> We replicate 4 popular LLM-based optimization code generation approaches [3-6] and directly compare SPA against them. We report the average improvements across all experimental models on HumanEval below. SPA not only outperforms these approaches in terms of Pass@1 improvements but also cost significantly less time.
>
> | Method | ΔPass@1 (%) | Time (Sec) |
> |--------|-------|----------|
> | Self-Debugging [3] | +4.2 | 27.3 |
> | Self-Planning [4] | +3.9 | 21.6 |
> | ReAct [5] | +1.1 | 28.8 |
> | Self-Edit [6] | +1.8 | 26.4 |
> | **SPA** | +**5.5** | **15.4** |
>
> \* *Self-debugging leverages error messages from test cases, while SPA doesn’t require any test case.*
>
> [3] Chen, X., Lin, M., Schärli, N., & Zhou, D. (2023, October 5). Teaching large language models to self-debug. In The Twelfth International Conference on Learning Representations.
>
> [4] Xue Jiang, Yihong Dong, Lecheng Wang, Zheng Fang, Qiwei Shang, Ge Li, Zhi Jin, and Wenpin Jiao. 2024. Self-Planning Code Generation with Large Language Models. ACM Trans. Softw. Eng. Methodol. 33, 7, Article 182 (September 2024), 30 pages. https://doi.org/10.1145/3672456
>
> [5] Yao, S., Zhao, J., Yu, D., Du, N., Shafran, I., Narasimhan, K., & Cao, Y. (2023, March 10). React: Synergizing reasoning and acting in language models. In The Eleventh International Conference on Learning Representations.
>
> [6] Zhang, K., Li, Z., Li, J., Li, G., & Jin, Z. (2023, July). Self-Edit: Fault-Aware Code Editor for Code Generation. In Proceedings of the 61st Annual Meeting of the Association for Computational Linguistics.

---

> ### Author Response · Authors · 2024-11-19
> **Response to Reviewer 3 (Part 2)**
>
> ---
>
> # Question 2 (Weakness 2)
>
> As the model generates different tokens, its attention dynamically changes at each step. Precisely locating the "most informative" tokens at all steps is extremely challenging. A recent study [7] has shown that attention can be overly distributed to the first or special tokens (a phenomenon called "attention sink"). Furthermore, determining how the model distributes its attention to sub-tokens is complex. Overly micro-managing specific tokens can easily lead to poor performance. For instance, if we incorrectly steer the model's attention to the wrong words in just 5% of cases, the final generated code may be incorrect. Therefore, in our approach, we pursue a balanced strategy. SPA anchors the natural language (NL) instruction in the code generation prompt. We chose this method because, although it may slightly reduce precision, the NL instruction remains consistently relevant to all generated code tokens. Thus, negative influence of less relevant tokens in the prompt can be counteracted by most other tokens. We promise to add this discussion in paper.
>
> In this work, we evaluate anchoring different components in code generation prompt in Section 5.4. We created four experiment baselines by not anchoring test cases and code. Our results indicate that "anchoring NL instruction alone in the prompt" achieves the best performance.
>
> | Anchored Text | HumanEval | HumanEval+ | MBPP | MBPP+ |
> |--------------|-----------|------------|------|--------|
> | *NL* | +**5.48** | +**5.08** | +**4.26** | +**3.22** |
> | *NL + Test* | +5.11 | +4.89 | +4.05 | +3.11 |
> | *NL + Code* | +4.87 | +4.65 | N/A | N/A |
> | *NL + Code + Test* | +4.76 | +4.57 | N/A | N/A |
>
> [7] Xiao, G., Tian, Y., Chen, B., Han, S., & Lewis, M. (2024, April 7). *Efficient streaming language models with attention sinks*. In The Twelfth International Conference on Learning Representations.
>
>
> ---
>
> # Question 3
>
> Thanks for the reviewer’s suggestion. This is an interesting question! We conducted additional experiments to study the number of tasks that SPA compromises initially correct output V.S. number of tasks that SPA rectifies initially incorrect output. Below are the average results for each model on HumanEval.
>
> | Model | Compromised | Rectified |
> |-------|-------------|-----------|
> | Codegen-350M | 5.7% | 9.5% |
> | DeepSeek-Coder-1.3B | 0.8% | 5.5% |
> | DeepSeek-Coder-6.7B | 12.2% | 17.6% |
> | CodeLlama-7B | 5.4% | 10.8% |
> | DeepSeek-Coder-33B | 0.8% | 1.6% |
>
> The overall ratio of "compromised" to "rectified" tasks is approximately 1:2. We find this result interesting. While SPA successfully corrects many incorrect generations, it also compromises some initially correct generated code. This result suggests that the current significant improvements of SPA can be further enhanced. For example, we could use test cases to decide when to trigger SPA (when any test case fails).
>
> To validate this design, we conducted additional experiments testing a setting where SPA is triggered only when the code fails to pass the test cases. The result shows that SPA serves effectively as an error correction approach, significantly improving Pass@1 (%) performance in this scenario. We will include these new results and design details in the paper.
>
> | Model | Original | +SPA (triggered on all tasks) | +SPA (triggered when test case failed) |
> |-------|----------|-----------------------------|------------------------------------|
> | Codegen-350M | 15.3% | 18.3% (+3.0%) | 20.1% (+4.8%) |
> | DeepSeek-Coder-1.3B | 66.4% | 69.5% (+3.1%) | 73.1% (+6.7%) |
> | DeepSeek-Coder-6.7B | 75.6% | 83.2% (+7.6%) | 88.5% (+12.9%) |
> | CodeLlama-7B | 33.6% | 40.5% (+6.9%) | 44.0% (+10.4%) |
> | DeepSeek-Coder-33B | 81.7% | 84.7% (+3.0%) | 86.2% (+4.5%) |

---

> ### Comment · Reviewer_wsgt · 2024-11-22
>
> Thanks for your reply. The author's answers to question 2 and weakness 2 are not convincing. I'll maintain the score.

---

> > ### Author Response · Authors · 2024-11-23
> > **Clarification on Question 2 and Weakness 2**
> >
> > Thank you for your response. We would be really grateful if you could elaborate on which part of our response is not convincing or if you could share any suggestions. In the meantime, we are running experiments and collecting quantitative evidence for our response to Question 2 and Weakness 2. We will post our results in the next couple of days. Thanks!

---

> > ### Author Response · Authors · 2024-11-24
> > **Additional experiments on "identifying and anchoring the most informative tokens" (Question 2/Weakness 2)**
> >
> > To investigate whether narrowing down the anchored text to informative tokens could improve performance, we conducted additional experiments using informative tokens labeled by human programmers as anchored tokens. Specifically, we made use of the dataset from Kou et al. [1], in which multiple human programmers manually annotated important tokens that a model needs to attend to when solving a programming task in HumanEval. Similar to the previous settings, we also tuned the attention weight hyperparameter using 20% of randomly sampled data.
> >
> > [1] Bonan Kou, Shengmai Chen, Zhijie Wang, Lei Ma, and Tianyi Zhang. 2024. Do Large Language Models Pay Similar Attention Like Human Programmers When Generating Code? Proc. ACM Softw. Eng. 1, FSE, Article 100 (July 2024), 24 pages. https://doi.org/10.1145/3660807
> >
> >
> > | Model | Original | +SPA (entire task description) | +SPA (informative tokens labeled by human programmers) |
> > |-------|----------|------------------------------|--------------------------------------------------|
> > | Codegen-350M-mono | 15.3 | 18.3 | 15.9 |
> > | Deepseek-coder-1.3b-instruct | 66.4 | 69.5 | 68.9 |
> > | Deepseek-coder-6.7b-instruct | 75.6 | 83.2 | 81.1 |
> > | CodeLlama-7b-hf | 33.6 | 40.5 | 39.0 |
> > | Deepseek-coder-33b-instruct | 81.7 | 84.7 | 82.9 |
> >
> >
> >
> > The table below shows the results. We found that while anchoring on human-labeled informative tokens improves pass@1 compared with the original code LLMs, it performs worse than anchoring on the entire NL task description.
> > We think there are two plausible reasons. First, since LLMs need to attend to different context tokens at each decoding step, providing a narrow set of anchored tokens may have a negative impact and distract the LLM in certain decoding steps. Second, previous studies such as [2] show that even though some tokens, such as separators and empty space, may not be semantically meaningful or informative, they provide important signals for LLMs to generate the right content (e.g., following the grammar rules). Thus, over-attending the informative tokens but not the special tokens in the task description may disrupt the regular generation process.
> >
> > [2] Xiao, G., Tian, Y., Chen, B., Han, S., & Lewis, M. (2024, April 7). Efficient streaming language models with attention sinks. In The Twelfth International Conference on Learning Representations.
> >
> > Nevertheless, we think this is a challenging but interesting future direction to investigate. We believe our findings will open up new research opportunities for the community on this topic.

---

### Official Review · Reviewer_daVy · 2024-11-03

**Soundness:** 3
**Presentation:** 3
**Contribution:** 3
**Rating:** 6
**Confidence:** 4

**Summary:**

The paper proposes SPA, an approach to improve code generation in LLMs by addressing attention dilution, where models lose focus on the initial prompt during extended generation. The authors demonstrate the limitations of current LLMs in maintaining prompt relevance over generated sequences, potentially leading to inaccuracies. SPA amplifies the attention on selected prompt tokens, significantly improving performance across different LLMs and benchmarks.

**Strengths:**

+ a new approach to improve LLMs for code generation
+ bring significant performance improvement

**Weaknesses:**

- generalizability is limited to code generation with Python
- missing prompt optimization-based baselines
- the randomness in the process of tuning anchoring strength should be explored
- the impact of prompt length is unknown

**Questions:**

1. In Table 1, some SPA-optimal models perform slightly behind the SPA-turned models, for example, pass@10 of DeepSeek-Coder on HumanEval+ and MBPP. Are there any reasons for this?

2. The authors used 15% of sampled data from a benchmark to tune the anchoring strength. If picking the 15% data was a random process, how stable is SPA with different sample sets?

3. As shown in Figure 4, the examined LLMs share similar trends regarding the performance of different anchoring strength values on the same dataset, is it possible to share/reuse anchoring strength among different LLMs?

4. Many existing studies optimize the prompts to make LLMs more focused on the key information of the provided prompts. Are there specific reasons that authors do not compare SPA to these prompt-optimization-based approaches?

5. Does SPA perform significantly differently on short prompts compared to long prompts?

---

> ### Author Response · Authors · 2024-11-18
> **Response to Reviewer 2 (Part 1)**
>
> Thanks for your insightful comments! We reply to each weakness and question below.
>
> ---
>
> # Weakness 1
>
> Thank you for your suggestion. We have conducted additional experiments on HumanEval-X [1] to show the generalizability of SPA to other programming languages. We will add the results in paper.
>
> | Model | Python | Java | JavaScript | C++ | Go |
> |-------|---------|------|------------|-----|-----|
> | Codegen-350M | 15.3% | 9.8% | 13.4% | 9.8% | 6.7% |
> | +SPA | 18.3% | 11.6% | 15.9% | 12.2% | 11.0% |
> | DeepSeek-Coder-1.3B | 66.4% | 42.7% | 57.3% | 43.3% | 40.2% |
> | +SPA | 69.5% | 45.1% | 59.8% | 45.1% | 42.1% |
> | DeepSeek-Coder-6.7B | 75.6% | 48.8% | 65.2% | 49.4% | 45.7% |
> | +SPA | 83.2% | 53.7% | 72.0% | 50.0% | 50.0% |
> | CodeLlama-7B | 33.6% | 22.0% | 29.3% | 22.0% | 20.1% |
> | +SPA | 40.5% | 26.2% | 34.8% | 26.2% | 24.4% |
> | DeepSeek-Coder-33B | 81.7% | 53.0% | 70.7% | 53.7% | 49.4% |
> | +SPA | 84.7% | 54.9% | 73.2% | 55.5% | 51.2% |
>
> [1] Qinkai Zheng, Xiao Xia, Xu Zou, Yuxiao Dong, Shan Wang, Yufei Xue, Lei Shen, Zihan Wang, Andi Wang, Yang Li, Teng Su, Zhilin Yang, and Jie Tang. 2023. CodeGeeX: A Pre-Trained Model for Code Generation with Multilingual Benchmarking on HumanEval-X. In Proceedings of the 29th ACM SIGKDD Conference on Knowledge Discovery and Data Mining (KDD '23). Association for Computing Machinery, New York, NY, USA, 5673–5684. https://doi.org/10.1145/3580305.3599790
>
> ---
>
> # Question 1
> This is because we SPA is tuned based on Pass@1, whose optimal values may slightly differ from the optimal values of Pass@10. We promise to clarify this in the paper.
>
> ---
>
> # Question 2 (Weakness 3)
> To demonstrate the tuning stability, we evenly split the original dataset into 5 subsets. Below we show the hyper-parameters tuned on each subset as well as the hyper-parameters tuned on the entire dataset. As demonstrated in Figure 4, the tuned hyper-parameters are near the optimal ones. It is stable because the hyperparameter-performance distribution follows a relatively simple unimodal pattern. We promise to better clarify this in paper.
>
>
>
> | Model | Subset | HumanEval/HumanEval+ | MBPP/MBPP+ |
> |-------|--------|---------------------|------------|
> | CodeGen-Mono-350M | Subset1 | 1.05 | 1.30 |
> | | Subset2 | 1.10 | 1.35 |
> | | Subset3 | 1.20 | 1.25 |
> | | Subset4 | 1.3 | 1.35 |
> | | Subset5 | 1.25 | 1.35 |
> | | Full | 1.20 | 1.35 |
> | DeepSeek-Coder-1.3B | Subset1 | 1.05 | 1.20 |
> | | Subset2 | 1.05 | 1.15 |
> | | Subset3 | 1.10 | 1.15 |
> | | Subset4 | 1.00 | 1.20 |
> | | Subset5 | 1.05 | 1.25 |
> | | Full | 1.05 | 1.20 |
> | DeepSeek-Coder-6.7B | Subset1 | 1.30 | 1.30 |
> | | Subset2 | 1.25 | 1.20 |
> | | Subset3 | 1.30 | 1.25 |
> | | Subset4 | 1.20 | 1.20 |
> | | Subset5 | 1.35 | 1.25 |
> | | Full | 1.28 | 1.25 |
> | CodeLlama-7B | Subset1 | 1.55 | 1.25 |
> | | Subset2 | 1.55 | 1.20 |
> | | Subset3 | 1.50 | 1.20 |
> | | Subset4 | 1.65 | 1.25 |
> | | Subset5 | 1.65 | 1.15 |
> | | Full | 1.60 | 1.20 |
> | DeepSeek-Coder-33B | Subset1 | 1.25 | 1.25 |
> | | Subset2 | 1.30 | 1.30 |
> | | Subset3 | 1.40 | 1.30 |
> | | Subset4 | 1.35 | 1.40 |
> | | Subset5 | 1.35 | 1.20 |
> | | Full | 1.35 | 1.30 |
>
> ---
>
> # Question 3
>
> We have conducted cross-model experiments in Section 5.2. The results in Table 2 demonstrate that the anchoring strength tuned on one model can be effectively transferred to another.
>
> Furthermore, we've found that setting a universal anchoring strength can enhance performance across all models (as detailed in Section 5.3 and Appendix A.3). For example, when we set the default value to 1.2, we observe consistent performance improvements as shown below. We promise to better clarify this in paper.
>
> | Model | HumanEval | HumanEval+ | MBPP | MBPP+ |
> |-------|-----------|------------|------|--------|
> | CodeGen-Mono-350M | 15.3 | 12.2 | 19.6 | 15.9 |
> | +SPA_default | 16.8 (+1.5) | 13.0 (+0.8) | 23.7 (+4.1) | 19.7 (+3.8) |
> | DeepSeek-Coder-1.3B | 66.4 | 61.8 | 58.2 | 52.4 |
> | +SPA_default | 71.0 (+4.6) | 65.3 (+3.5) | 61.7 (+3.5) | 53.2 (+0.8) |
> | DeepSeek-Coder-6.7B | 75.6 | 70.2 | 67.0 | 58.5 |
> | +SPA_default | 81.9 (+6.3) | 74.7 (+4.5) | 69.6 (+2.6) | 59.8 (+1.3) |
> | CodeLlama-7B | 33.6 | 28.2 | 50.9 | 40.8 |
> | +SPA_default | 34.6 (+1.0) | 29.2(+1.3) | 52.7 (+1.8) | 43.0 (+2.2) |
> | DeepSeek-Coder-33B | 81.7 | 77.1 | 73.4 | 63.2 |
> | +SPA_default | 82.7 (+1.0) | 77.2 (+0.1) | 75.4 (+2.0) | 66.0 (+2.7) |

---

> ### Author Response · Authors · 2024-11-18
> **Response to reviewer 2 (Part 2)**
>
> ---
>
> # Question 4 (Weakness 2)
>
> We replicate popular prompt optimization-based code generation approaches [2-5] and directly compare SPA against them. We report the average improvements across all experimental models on HumanEval below. SPA not only outperforms these prompt-based approaches in terms of Pass@1 improvements but also cost significantly less time.
>
>
> | Method | ΔPass@1 (%) | Time (Sec) |
> |--------|------------------------|------------|
> | Self-Debugging [2] | +4.2 | 27.3 |
> | Self-Planning [3] | +3.9 | 21.6 |
> | ReAct [4] | +1.1 | 28.8 |
> | Self-Edit [5] | +1.8 | 26.4 |
> | **SPA** | +**5.5** | **15.4** |
>
> \* *Self-debugging leverages error messages from test cases, while SPA doesn’t require any test case.*
>
> Notably, compared to most prompt-optimization approaches, SPA enhances code generation performance at the logits level, which can be easily integrated with them to form a more advanced pipeline. We will add the results and discussion in the paper.
>
> [2] Chen, X., Lin, M., Schärli, N., & Zhou, D. (2023, October 5). Teaching large language models to self-debug. In The Twelfth International Conference on Learning Representations.
>
> [3] Xue Jiang, Yihong Dong, Lecheng Wang, Zheng Fang, Qiwei Shang, Ge Li, Zhi Jin, and Wenpin Jiao. 2024. Self-Planning Code Generation with Large Language Models. ACM Trans. Softw. Eng. Methodol. 33, 7, Article 182 (September 2024), 30 pages. https://doi.org/10.1145/3672456
>
> [4] Yao, S., Zhao, J., Yu, D., Du, N., Shafran, I., Narasimhan, K., & Cao, Y. (2023, March 10). React: Synergizing reasoning and acting in language models. In The Eleventh International Conference on Learning Representations.
>
> [5] Zhang, K., Li, Z., Li, J., Li, G., & Jin, Z. (2023, July). Self-Edit: Fault-Aware Code Editor for Code Generation. In Proceedings of the 61st Annual Meeting of the Association for Computational Linguistics.
>
>
> ---
>
> # Question 5 (Weakness 4)
>
> We appreciate the reviewer's insightful question about prompt length.
> We divided the HumanEval dataset into three subsets (Short, Medium, and Long) based on the 33rd and 66th percentiles of prompt lengths (Average length is 451 characters). The effectiveness of SPA across these divisions is as follows:
>
> | Model | Short | Medium | Long |
> |-------|-------|---------|------|
> | CodeGen-Mono-350M | 37.02% | 6.6% | 2.3% |
> | +SPA | 38.72% (+1.7%) | 9.6% (+3.0%) | 6.6% (+4.3%) |
> | DeepSeek-Coder-1.3B | 81.8% | 45.5% | 29.6% |
> | +SPA | 80.0% (-1.8%) | 60.0% (+14.5%) | 55.6% (+26%) |
> | DeepSeek-Coder-6.7B | 87.3% | 69.1% | 44.4% |
> | +SPA | 90.9% (+3.6%) | 65.5% (-3.6%) | 51.9% (+7.5%) |
> | CodeLlama-7B | 69.2% | 43.5% | 0% |
> | +SPA | 71.8% (+2.6%) | 43.5% (+0%) | 10.0% (+10.0%) |
> | DeepSeek-Coder-33B | 85.5% | 81.8% | 68.5% |
> | +SPA | 87.3% (+1.8%) | 81.8% (+0%) | 75.9% (+7.4%) |
>
> We find the results interesting. While LLMs consistently perform better on short prompts, **SPA is consistently more effective when handling longer prompts**. This finding confirms that SPA can effectively address attention dilution issue observed in our empirical study. It also suggests that SPA is particularly beneficial when dealing with lengthy prompts. We promise to include these new results and discussion in paper.

---

### Official Review · Reviewer_xwom · 2024-11-04

**Soundness:** 2
**Presentation:** 2
**Contribution:** 2
**Rating:** 5
**Confidence:** 4

**Summary:**

The authors propose a new method to improve the code generation quality of LLMs by enhancing the attention mechanism. To show the effectiveness of their method on HumanEval(+) and MBPP(+) datasets, they use 1/5 tasks in the datasets to set the hyperparameters and the other 4/5 tasks to evaluate the performance.

**Strengths:**

- The overall structure is clear and easy to follow.
- The authors conduct experiments on several open Code LLMs like CodeGen, DeepSeek-Coder and Code Llama with different model sizes.
- The authors further provide ablation studies to validate the effectiveness of their methods from different perspectives.

**Weaknesses:**

The contributions of the paper could be very limited, and the efficacy of the research is still questionable without further experiments.
Several weaknesses include:
- The motivation of the paper is not strong enough. For example, the "attention dillusion" phenomenon is not a new phenomenon and has been discussed in the literature [1-3]. It appears that the lack of attention is quite common in the existing LLMs, not just in code genaration. The authors should provide more motivation as to why they only focus on the code generation task.
- The efficacy of SPA is not very convincing. Essentially, SPA requires tuning/searching the hyperparameters (e.g., anchoring strength) on each benchmark, which makes it impractical.
- Although there is an ablation study on cross-dataset evaluation, it is still not enough to validate the effectiveness of SPA, as HumanEval and MBPP are in the same algorithmic paradigm. Widely-used open-domain code benchmarks like BigCodeBench [4] should be used to further validate the effectiveness of SPA.
- Section 5.3 shows that the anchoring weight affects the performance of SPA significantly, and results in various performance across models and datasets. It is unclear whether the proposed method can generalize without hyperparameter tuning.
- The evaluated Code LLMs are quite outdated. The authors should include more recent models like StarCoder2 [5] and DeepSeek-Coder-V2 [6].
- As pointed in A.5, SPA additionally takes 2 to 3.5 times longer than regular inference, with extra memory usage. This limitation further limits the usability of SPA.
- The current evaluation only focuses on Python-only and function-level code generation, and it is unclear how the proposed method can generalize to other programming languages and code generation tasks.


[1] Shi, F., Chen, X., Misra, K., Scales, N., Dohan, D., Chi, E. H., ... & Zhou, D. (2023, July). Large language models can be easily distracted by irrelevant context. In International Conference on Machine Learning (pp. 31210-31227). PMLR.

[2] Liu, N. F., Lin, K., Hewitt, J., Paranjape, A., Bevilacqua, M., Petroni, F., & Liang, P. (2024). Lost in the middle: How language models use long contexts. Transactions of the Association for Computational Linguistics, 12, 157-173.

[3] Zhang, Q., Singh, C., Liu, L., Liu, X., Yu, B., Gao, J., & Zhao, T. Tell Your Model Where to Attend: Post-hoc Attention Steering for LLMs. In The Twelfth International Conference on Learning Representations.

[4] Zhuo, T. Y., Vu, M. C., Chim, J., Hu, H., Yu, W., Widyasari, R., ... & Von Werra, L. (2024). Bigcodebench: Benchmarking code generation with diverse function calls and complex instructions. arXiv preprint arXiv:2406.15877.

[5] Lozhkov, A., Li, R., Allal, L. B., Cassano, F., Lamy-Poirier, J., Tazi, N., ... & de Vries, H. (2024). Starcoder 2 and the stack v2: The next generation. arXiv preprint arXiv:2402.19173.

[6] Zhu, Q., Guo, D., Shao, Z., Yang, D., Wang, P., Xu, R., ... & Liang, W. (2024). DeepSeek-Coder-V2: Breaking the Barrier of Closed-Source Models in Code Intelligence. arXiv preprint arXiv:2406.11931.

**Questions:**

See the weaknesses section.

---

> ### Author Response · Authors · 2024-11-18
> **Response to Reviewer 1 (Part 1)**
>
> Thanks for your insightful comments! We reply to each weakness below.
>
> ---
>
> # Weakness 1
>
> Thank you for mentioning these works. They are related but the phenomena described in these works are different from the phenomena in our findings. Specifically, the first work [1] found that LLMs can be distracted by irrelevant information. However, in our work, the code generation prompts do not include any irrelevant information. The function header and natural language instruction are consistently relevant to the generated code. The second work [2] found that LLMs struggle to attend to the middle of long contexts. However, code generation prompts are usually short and appear at the beginning. The third work [3] points out that model attention should align with human intention in human-AI communication. However, code generation prompts directly represent user intentions in our experiments.
> Our work is the first to confirm the existence of an attention dilution issue in code generation tasks.
>
> It's also worth noting that the first and second papers [1, 2] are empirical studies and didn't propose any technical solutions. The third paper [3] proposes a technical approach that requires user input to steer model attention. In contrast, SPA automatically amplifies the influence of original prompt to address attention dilution issue and doesn't require any user input.
> Furthermore, it requires extensive model profiling to identify attention headers to be adjusted for improvement. During inference, it needs to recalculate attention distribution for each layer and selected headers. In contrast, SPA doesn't require any model profiling and can directly adjust attention based on logits difference.
>
>
> We will cite these papers recommended by the reviewer and clarify the differences in the paper.
>
> [1] Shi, F., Chen, X., Misra, K., Scales, N., Dohan, D., Chi, E. H., ... & Zhou, D. (2023, July). Large language models can be easily distracted by irrelevant context. In International Conference on Machine Learning (pp. 31210-31227). PMLR.
>
> [2] Liu, N. F., Lin, K., Hewitt, J., Paranjape, A., Bevilacqua, M., Petroni, F., & Liang, P. (2024). Lost in the middle: How language models use long contexts. Transactions of the Association for Computational Linguistics, 12, 157-173.
>
> [3] Zhang, Q., Singh, C., Liu, L., Liu, X., Yu, B., Gao, J., & Zhao, T. Tell Your Model Where to Attend: Post-hoc Attention Steering for LLMs. In The Twelfth International Conference on Learning Representations.
>
> ---
>
> # Weakness 2
>
> Unlike approaches requiring multiple hyperparameter tuning, SPA only needs to tune one hyperparameter (anchoring strength) for optimal performance. This hyperparameter follows a simple pattern (Figure 4) and is easy to tune. In practice, we can set a default value of 1.2, which improves performance across various models and benchmarks. Please see the results in our response to Weakness 4.
>
>
> Additionally, our approach is model-agnostic. It can be applied to other architectures. In contrast, the existing attention steering method such as [3] requires a model profiling stage, which is not only model-specific but time-consuming in practice.
>
> ---

---

> ### Author Response · Authors · 2024-11-19
> **Response to reviewer 1 (Part 2)**
>
> # Weakness 3
> Following the reviewer’s suggestion, we've conducted additional experiments on BigCodeBench. Please check the results below. While the absolute improvements aren't as big as in simple benchmarks, the relative improvements remain comparable. For example, although the absolute improvement for CodeGen-Mono-350M is 0.3%, SPA enhances its performance by 27% relative to the original 1.1% performance. This is because SPA only adjusts the attention of the code generation model and therefore still relies on the model's innate capability of code generation. In other words, if a model could solve a task but misses a few tokens or requirements in the prompt, SPA can help with this by adjusting the attention. If a model is very poor and doesn't possess the capability to solve a task, adjusting the model attention won't help much. We promise to include these new results and discussion in the paper.
>
> | Model | BigCodeBench | HumanEval | HumanEval+ | MBPP | MBPP+ |
> |-------|--------------|-----------|------------|------|--------|
> | CodeGen-Mono-350M | 1.1 | 15.3 | 12.2 | 19.6 | 15.9 |
> | +SPA | 1.4 (+0.3) (27%) | 18.3 (+3.0) (20%) | 16.0 (+3.8) (31%) | 24.9 (+5.3) (27%) | 20.6 (+4.7) (30%) |
> | DeepSeek-Coder-1.3B | 2.5 | 66.4 | 61.8 | 58.2 | 52.4 |
> | +SPA | 3.3 (+0.8) (32%) | 69.5 (+3.1) (5%) | 66.4 (+4.6) (7%) | 59.1 (+0.9) (2%) | 52.4 (+0.0) (0%) |
> | DeepSeek-Coder-6.7B | 12.7 | 75.6 | 70.2 | 67.0 | 58.5 |
> | +SPA | 14.2 (+1.5) (12%) | 83.2 (+7.6) (10%) | 75.6 (+5.4) (8%) | 69.6 (+2.6) (4%) | 60.2 (+1.7) (3%) |
> | CodeLlama-7B | 3.4 | 33.6 | 28.2 | 50.9 | 40.8 |
> | +SPA | 3.8 (+0.4) (12%) | 40.5 (+6.9) (21%) | 33.6 (+5.4) (19%) | 52.9 (+2.0) (4%) | 43.1 (+2.3) (6%) |
> | DeepSeek-Coder-33B | 18.9 | 81.7 | 77.1 | 73.4 | 63.2 |
> | +SPA | 20.7 (+1.8) (10%) | 84.7 (+3.0) (4%) | 77.9 (+0.8) (1%) | 77.2 (+3.8) (5%) | 68.5 (+5.3) (8%) |
>
>
> # Weakness 4
> Yes!
> In practice, we can set a default value of 1.2 (Section 5.3 & Appendix A.3), which improves performance across various models and benchmarks.
> Below we shows Pass@1 rates when the anchoring effect is set to the default 1.2.
> We will include this result in paper.
>
> | Model | HumanEval | HumanEval+ | MBPP | MBPP+ |
> |-------|-----------|------------|------|--------|
> | CodeGen-Mono-350M | 15.3 | 12.2 | 19.6 | 15.9 |
> | +SPA_default | 16.8 (+1.5) | 13.0 (+0.8) | 23.7 (+4.1) | 19.7 (+3.8) |
> | DeepSeek-Coder-1.3B | 66.4 | 61.8 | 58.2 | 52.4 |
> | +SPA_default | 71.0 (+4.6) | 65.3 (+3.5) | 61.7 (+3.5) | 53.2 (+0.8) |
> | DeepSeek-Coder-6.7B | 75.6 | 70.2 | 67.0 | 58.5 |
> | +SPA_default | 81.9 (+6.3) | 74.7 (+4.5) | 69.6 (+2.6) | 59.8 (+1.3) |
> | CodeLlama-7B | 33.6 | 28.2 | 50.9 | 40.8 |
> | +SPA_default | 34.6 (+1.0) | 29.2(+1.3) | 52.7 (+1.8) | 43.0 (+2.2) |
> | DeepSeek-Coder-33B | 81.7 | 77.1 | 73.4 | 63.2 |
> | +SPA_default | 82.7 (+1.0) | 77.2 (+0.1) | 75.4 (+2.0) | 66.0 (+2.7) |
>
> # Weakness 5
>
> Following the reviewer’s suggestion, we've conducted additional experiments on DeepSeek-Coder-V2 (16B) and StarCoder2 (15B). Please see the results in the table below. It shows can SPA can consistently improve recent code LLMs.
>
> | Model | HumanEval | HumanEval+ | MBPP | MBPP+ | BigCodeBench |
> |-------|-----------|------------|------|--------|--------------|
> | DeepSeek-Coder-V2-16B | 85.4 | 82.3 | 89.4 | 75.1 | 17.2 |
> | +SPA | 88.4 (+3.0) | 83.7 (+1.4) | 92.1 (+2.7) | 76.7 (+1.6) | 19.0 (+1.8) |
> | StarCoder2 | 67.7 | 60.4 | 78.0 | 65.1 | 13.3 |
> | +SPA | 72.1 (+4.4) | 63.6 (+3.2) | 80.9 (+2.9) | 67.6 (+2.5) | 14.1 (+0.8) |

---

> ### Author Response · Authors · 2024-11-20
> **Response to Reviewer 1 (Part 3)**
>
> ---
>
> # Weakness 6
>
> In practice, we argue that the additional overhead is negligible for developers. On average, SPA took 15.4 seconds to complete a HumanEval task, which is comparable to the original model's 9.6 seconds. During the rebuttal, we've also integrated KV-cache and flash attention to speed up SPA. The overhead is reduced to only 1.6 times that of the original model.
> To illustrate, we report the decoding speed with and without SPA on our machine.
>
> | Model | Token/Second |
> |-------|--------------|
> | Codegen-350M | 34.1 |
> | +SPA | 23 .5|
> | DeepSeek-Coder-1.3B | 17.8 |
> | +SPA | 11.1 |
> | DeepSeek-Coder-6.7B | 12.1 |
> | +SPA | 7.6 |
> | CodeLlama-7B | 14.5 |
> | +SPA | 9.2 |
> | DeepSeek-Coder-33B | 5.3 |
> | +SPA | 3.3 |
>
> We will provide the implementation details and more discussion about the inference time in appendix.
>
>
>
> ---
>
> # Weakness 7
>
> Thank you for your suggestion. In addition to BigCodeBench which includes Multiple-E [5], we have also experimented with HumanEval-X [6] to show the generalizability of our approach to other programming languages. We will add the results in paper.
>
> | Model | Python | Java | JavaScript | C++ | Go |
> |-------|---------|------|------------|-----|-----|
> | Codegen-350M | 15.3% | 9.8% | 13.4% | 9.8% | 6.7% |
> | +SPA | 18.3% | 11.6% | 15.9% | 12.2% | 11.0% |
> | DeepSeek-Coder-1.3B | 66.4% | 42.7% | 57.3% | 43.3% | 40.2% |
> | +SPA | 69.5% | 45.1% | 59.8% | 45.1% | 42.1% |
> | DeepSeek-Coder-6.7B | 75.6% | 48.8% | 65.2% | 49.4% | 45.7% |
> | +SPA | 83.2% | 53.7% | 72.0% | 50.0% | 50.0% |
> | CodeLlama-7B | 33.6% | 22.0% | 29.3% | 22.0% | 20.1% |
> | +SPA | 40.5% | 26.2% | 34.8% | 26.2% | 24.4% |
> | DeepSeek-Coder-33B | 81.7% | 53.0% | 70.7% | 53.7% | 49.4% |
> | +SPA | 84.7% | 54.9% | 73.2% | 55.5% | 51.2% |
>
>
> [5] Cassano, F., Gouwar, J., Nguyen, D., Nguyen, S., Phipps-Costin, L., Pinckney, D., Yee, M.-H., Zi, Y., Anderson, C. J., Feldman, M. Q., Guha, A., Greenberg, M., & Jangda, A. (2022, December 19). MULTIPL-E: A scalable and extensible approach to benchmarking neural code generation. arXiv.org. https://arxiv.org/abs/2208.08227
>
> [6] Qinkai Zheng, Xiao Xia, Xu Zou, Yuxiao Dong, Shan Wang, Yufei Xue, Lei Shen, Zihan Wang, Andi Wang, Yang Li, Teng Su, Zhilin Yang, and Jie Tang. 2023. CodeGeeX: A Pre-Trained Model for Code Generation with Multilingual Benchmarking on HumanEval-X. In Proceedings of the 29th ACM SIGKDD Conference on Knowledge Discovery and Data Mining (KDD '23). Association for Computing Machinery, New York, NY, USA, 5673–5684. https://doi.org/10.1145/3580305.3599790

---

> > ### Comment · Reviewer_xwom · 2024-12-03
> >
> > Dear author, thank you for your reply. Your reply clarified some of my doubts. I increase the score to 5 points.

---

### Meta-Review · Area_Chair_CTR9 · 2024-12-19

**Metareview:**

I read the paper, reviewer comments and author rebuttal. This paper was borderline and overall I feel can be improved by another round of reviews. Please consider some suggestions below for future submission:

- The paper currently feels very specific to couple of benchmarks and it is not clear if the results are generalizable. Authors kindly did more experiments with BigBench in the rebuttal period but found much less improvement compared to the two main benchmarks in the paper. This is especially concerning with the additional overhead required in using the approach for each coding task.

- It is not clearly justified why the studied issue of attention dilution is specific to code generation settings. It is also not sure whether this phenomenon is only specific to older class of models.

- The gap compared to existing prompt optimization baselines is limited while the proposed approach requires extra information (full logits) and cannot work with closed source LLMs unlike baselines like ReAct etc.

**Additional Comments On Reviewer Discussion:**

Reviewers had concerns about generalizability beyond two specific benchmarks, limited discussion of hyperparameters and comparison to baselines. Authors kindly replied in the rebuttal but their response doesn't seem convincing.

---

### Decision · Program_Chairs · 2025-01-22

Reject